# *Compress to Impress*: Efficient LLM Adaptation Using a Single Gradient Step on 100 Samples

**Shiva Sreeram**[1]* **Alaa Maalouf**[2,1] **Pratyusha Sharma**[1] **Daniela Rus**[1]

[1]MIT CSAIL  [2]University of Haifa

## Abstract

Recently, Sharma et al. suggested a method called LAyer- SElective-Rank reduction (LASER) which demonstrated that pruning high-order components of carefully chosen LLM's weight matrices can boost downstream accuracy—without any gradient-based fine-tuning. Yet LASER's exhaustive, per-matrix search (each requiring full-dataset forward passes) makes it impractical for rapid deployment. We demonstrate that this overhead can be removed and find that: (i) Only a small, carefully chosen subset of matrices needs to be inspected—eliminating the layer-by-layer sweep, (ii) The gradient of each matrix's singular values pinpoints which matrices merit reduction, (iii) Increasing the factorization search space by allowing matrices rows to cluster around multiple subspaces and then decomposing each cluster separately further reduces overfitting on the original training data and further lifts accuracy by up to 24.6 percentage points, and finally, (iv) we discover that evaluating on just 100 samples rather than the full training data—both for computing the indicative gradients and for measuring the final accuracy—suffices to further reduce the search time; we explain that as adaptation to downstream tasks is dominated by prompting style, not dataset size. As a result, we show that combining these findings yields a fast and robust adaptation algorithm for downstream tasks. Overall, with a single gradient step on 100 examples and a quick scan of the top candidate layers and factorization techniques, we can adapt LLMs to new datasets—entirely without fine-tuning.

## 1 Introduction

Transformer-based large language models (LLMs) have rapidly become the backbone of modern natural-language systems, scaling from hundreds of millions to billions or trillions of parameters and achieving remarkable zero-shot and few-shot performance across a wide range of tasks Brown et al. [2020], Touvron et al. [2023]. Despite their success, adapting these models to domain-specific data remains expensive: standard fine-tuning requires back-propagation through all parameters, large GPU clusters, and hundreds of gradient steps. Even parameter-efficient methods such as LoRA Hu et al. [2022] and prompt-tuning Lester et al. [2021] still incur non-negligible compute and storage overhead when multiple tasks or domains must be supported simultaneously.

A complementary approach is to explore post-training interventions that modify a pretrained model without gradient-based optimization. Lately, LAyer-SElective-Rank reduction (LASER) of Sharma et al. provided a striking result: simply pruning higher-order components of carefully chosen weight matrices can increase downstream accuracy—no additional data, optimizers, or training epochs required. Unfortunately, LASER's exhaustive, per-matrix search demands a large-dataset forward pass per matrix in every layer, making it impractical for rapid deployment or on-device adaptation.

**Our contribution.** In this paper we revisit LASER through the lens of efficiency. Our key insight is that the matrices most responsible for task adaptation can be identified without an exhaustive sweep. By computing the gradient of each matrix's singular values, without ever updating model weights, on a small validation subset (100 data points only each computed just once) and allowing rows

---

*Correspondence: `sasreera@mit.edu`

39th Conference on Neural Information Processing Systems (NeurIPS 2025).

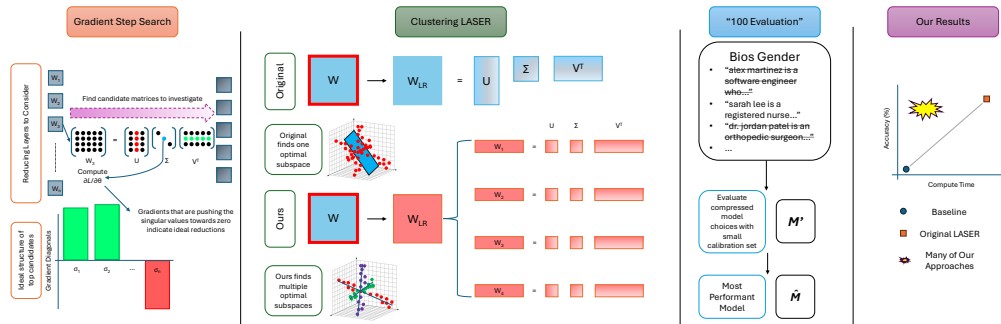

Figure 1: **Efficient LLM adaptation.** We present a method to adapt LLMs to new styles/domains without fine-tuning. (1) With a single gradient step on the target data, we compute gradients of the singular values across all weight matrices; these gradients rank which matrices merit low-rank compression to curb overfitting and align to the new style. (2) We broaden the search by clustering the rows of the selected matrices into multiple (best fitting) subspaces and factor each cluster, capturing heterogeneous structure and reducing noise/overfitting that manifest differently across row groups. (3) We show that both the gradient scoring and evaluation can be done with just 100 examples. (4) Finally, this yields up to 52× speedups and up to +24.6-point accuracy gains—no fine-tuning required.

to be decomposed around multiple subspaces, we both narrow the layers search space and unlock richer factorizations that further reduce overfitting. Guided by these observations, we develop a fast, sample-efficient adaptation algorithm that needs only a single gradient step on roughly 100 examples and a quick scan of a handful of candidate matrices.

Our contributions offers three key findings and a complete algorithm they enable:

1. **Gradient-guided matrix selection.** We show that the gradient of singular values reliably pinpoints which weight matrices merit reduction, eliminating the layer-by-layer sweep required by LASER.

2. **Sample-efficient evaluation.** We demonstrate that 100 labeled examples suffice for both gradient estimation and accuracy checks, i.e., can be used as the full given training data, indicating that adaptation quality is dominated by prompting style rather than dataset size.

3. **Multi-subspace factorization.** Clustering matrix rows into several subspaces and then performing rank reduction within each cluster enlarges the factorization search space and thus further mitigates overfitting, raising benchmark accuracy by up to 24.6 percentage points.

4. **Adapting LLMs.** Taken together, these findings yield a lightweight, training-free pipeline for adapting pretrained LLMs to new domains efficiently on a single GPU.

In short, with just one gradient step on 100 examples—and a rapid check of the most promising layers and factorization schemes—LLMs can be adapted to new datasets without any fine-tuning. We hope these findings and this approach broadens the practical reach of LLMs—particularly in settings where compute, bandwidth, or labeled data are scarce.

## 2 Related work

To our knowledge, Sharma et al. were the first to show that *targeted* rank-reduction of weight matrices can improve LLMs accuracy on downstream-tasks. Nevertheless, three established research streams are highly relevant:(i) how large language models internally encode factual knowledge, (ii) how over-parameterized networks can be compressed without sacrificing performance, and (iii) how to adapt LLMs to down stream tasks.

**How facts are stored.** Early probing work [Ettinger et al., 2016, Adi et al., 2016, Hupkes et al., 2018, Conneau et al., 2018] suggests that factual attributes are distributed across layers. One influential hypothesis posits that entity-specific information is cached in two-layer key–value memories inside MLP blocks [Geva et al., 2021] and then propagated forward by self-attention [Elhage, 2021]. Evidence for this locality comes from interventions that locate and overwrite such memories to produce counter-factual responses [Meng et al., 2022], as well as from "early-exit" behaviour, where

intermediate representations alone suffice for correct generation [Zhao et al., 2021]. Conversely, Hase et al. [2023] show that editing multiple layers is required to alter answers involving overlapping entities, hinting at a more fragmented, cross-layer storage scheme. We do not adjudicate between these views; instead, we rely on the observation that high-rank components often act as noise and that retaining only low-rank structure can surface the correct answer.

**Aligning LLMs to downstream tasks.** Probing studies have shown that LLMs are not world models for any given task, necessitating rapid adaptation processes Qi et al. [2023], Zhao et al. [2025], Sreeram et al. [2025]. The earliest strategy for LLMs alignment was full supervised fine-tuning on task data [Radford et al., 2019, Brown et al., 2020], but computational cost motivated **parameter-efficient** techniques such as adapters [Houlsby et al., 2019], IA$^3$ [Liu et al., 2022a], prefix-tuning [Li and Liang, 2021], prompt-tuning [Lester et al., 2021], P-Tuning v2 [Liu et al., 2022b], and low-rank adaptation (LoRA) [Hu et al., 2022], all of which update $\ll 1\%$ of the weights. A complementary line of work, **instruction tuning**, aligns models via supervised fine-tuning on diverse (instruction, response) pairs, improving zero-shot generalization [Wei et al., Chung et al., 2024, Sanh et al., 2022]. Coverage can be expanded almost for free with synthetic data generators such as Self-Instruct [Wang et al., 2023] or Alpaca [Taori et al., 2023]. Learning based on LLM or VLM features has also become a growing trend for adapting these representations to downstream tasks Chahine et al. [2024], Maalouf et al. [2024], Wang et al. [2024, 2025]. Beyond supervised objectives, **reinforcement learning from human feedback** trains a reward model from pairwise preferences and optimizes it with RL [Stiennon et al., 2020, Ouyang et al., 2022, Ziegler et al., 2019]; recent variants like Direct Preference Optimization (DPO) [Rafailov et al., 2023] and SteerLM [Dong et al., 2023] replace unstable policy-gradient updates with simple classification or attribute-conditioned losses. At inference time, prompt engineering—including chain-of-thought and zero/few-shot prompting—offers a parameter-free alignment layer [Wei et al., 2022, Kojima et al., 2022].

**Compressing neural networks by pruning.** *Unstructured pruning* methods trim networks by zeroing individual weights while aiming to preserve each layer's output [LeCun et al., 1990]. Some embed sparsity into training via constraints or regularizers [Lebedev and Lempitsky, 2016, Dong et al., 2017a, Iandola et al., 2016, Aghasi et al., 2017, Lin et al., 2017]. Others prune *post hoc*, dropping weights below a magnitude threshold [Han et al., 2015, Renda et al., 2020, Guo et al., 2016]. Data-aware schemes rank weights using loss or activation statistics from a mini-batch [Baykal et al., 2019a,b, Gamboa et al., 2020, Lin et al., 2020, Molchanov et al., 2017, 2019, Yu et al., 2018]. For broader reviews, see [Gale et al., 2019, Blalock et al., 2020]. On the other hand, *Structured pruning* removes whole channels, neurons, or filters, these methods cut memory usage and speed up inference on any hardware [Li et al., 2019, Luo and Wu, 2018, Tukan et al., 2022a]. A wide range of strategies has been proposed [Liu et al., 2019b, Li et al., 2019, Chen et al., 2020, He et al., 2019, Dong et al., 2017b, Kang and Han, 2020, Ye et al., 2020, 2018], most of which assign each filter an importance score—either weight-based [He et al., 2017, 2018] or data-driven [Maalouf et al., 2021a, Liebenwein et al., 2020]—and prune those falling below a threshold. Notably, these methods aim to maintain the model's accuracy, typically by applying fine-tuning after compression. Many implementations iterate this prune-and-fine-tune cycle, incurring multiple costly retraining rounds [Renda et al., 2020]. Finally, we note that it is common in literature in this domain to have reductions in accuracy when improving runtime. This can be seen for papers that perform model compression [Baykal et al., 2019a], as well as for compressing datasets [Wang et al., 2018, Killamsetty et al., 2021].

**Low-rank approximations.** Layer compression can also be achieved by factorizing a heavy layer into several low-rank components Denton et al. [2014], Jaderberg et al. [2014], Maalouf et al. [2020, 2022], Kim et al. [2015], Tai et al. [2015], Ioannou et al. [2015], Alvarez and Salzmann [2017], Tukan et al. [2021], Yu et al. [2017], Lebedev et al. [2015], Liebenwein et al. [2021]. Complementary methods rely on weight sharing, random projections, or feature hashing to shrink parameter counts Weinberger et al. [2009], Tukan et al. [2021], Chen et al. [2015a,b], Ullrich et al. [2017]. Closest to our work, Maalouf et al. [2020] iteratively solve a projection-clustering objective to decompose LLM embedding layers. Liebenwein et al. [2021] extend this idea by distributing a size budget network-wide and applying multiple SVDs per layer, enabling whole-model compression. We build on the multi-subspace view, but re-propose to remove noise around each subspace rather than merely for reducing parameters.

## 3 Method

Our first goal is to identify *which* weight matrices in the given LLM should be compressed to improve its capabilities on a new dataset—without any gradient-based fine-tuning.

**Approach and Motivation.** Instead of attempting to compress all weights, the model first identifies which layers are most critical by analyzing gradients on a small calibration set. Key to our approach is the observation that the **gradient of each singular value** of a given matrix $W$ already tells us whether the model wishes to *shrink* or *expand* that component. If the loss pushes a singular value $\sigma_i$ toward *zero*, the corresponding rank-one direction does *not* contribute to the task or even harms, and can be removed; conversely, a positive push signals that it is useful and should be kept. We therefore rank matrices by the magnitude and sign of these *singular-value gradients* and apply low-rank decompositions only where the evidence for shrinkage is strongest. Within each chosen layer, weights are partitioned into blocks, and only the most informative directions are retained through a low-rank approximation; accuracy is computed with the calibration dataset to obtain the most valuable layer compression strategy.

---

**Algorithm 1:** BLOCK-FIRST GRADIENT LOW-RANK ADAPTATION

---

**Input:** LLM $\mathcal{M}$ with weights $\{W^\ell\}_{\ell=1}^L$; calibration set $\mathcal{D}$;
    row-clusters $K$; target block rank $j$; matrices to compress $q$.
**Output:** Compressed model $\widehat{\mathcal{M}}$
**1. Back-prop on $\mathcal{D}$: foreach** $(x, y) \in \mathcal{D}$ **do**
     $L \leftarrow \text{loss}(\mathcal{M}(x), y)$; back-prop; $G^\ell \mathrel{+}= \partial L / \partial W^\ell \ \forall \ell$
**2. Score matrices: foreach** $\ell = 1{:}L$ **do**
     partition $(W^\ell, G^\ell) \rightarrow (W_k, G_k)_{k=1}^K$; $s^\ell \leftarrow 0$; **foreach**
     $k = 1{:}K$ **do**
         $(U, \Sigma, V) = \text{thinSVD}(W_k)$; $\mathbf{g} = \text{diag}(U^\top G_k V)$;
         $s^\ell \mathrel{+}= -\sum_{i=r_k-j+1}^{r_k} \min(g_i, 0)$
     $s^\ell \leftarrow s^\ell / K$
$\mathcal{S} \leftarrow$ top-$q$ indices by $s^\ell$.
**3. Compress+Evaluate: foreach** $\ell \in \mathcal{S}$ *and* $(x, y) \in \mathcal{D}$ **do**
     reuse $(W_k)_1^K$; **foreach** $k$ **do**
         $(U, \Sigma, V) = \text{thinSVD}(W_k)$; keep top-$j$;
         $\widehat{W}_k = U \widehat{\Sigma} V^\top$
     $\widehat{W}^\ell = \text{stack}(\widehat{W}_k)_{k=1}^K$; $W^\ell \leftarrow \widehat{W}^\ell$
**return** $\widehat{\mathcal{M}}$

---

## 3.1 Gradient of a Singular Value w.r.t. the Loss

**Intuition.** Let $W = U \, \text{diag}(\sigma_1, \ldots, \sigma_r) \, V^\top \in \mathbb{R}^{m \times n}, r = \text{rank}(W)$, be the (thin) singular-value decomposition (SVD) of a weight matrix $W \in \mathbb{R}^{m \times n}$ that lives inside a given LLM model. After back-propagation we already have the ordinary matrix gradient $G := \partial L / \partial W \in \mathbb{R}^{m \times n}$. Infinitesimally perturbing the $i$-th singular value by $\text{d}\sigma_i$ changes the weight by $\text{d}W = u_i v_i^\top \, \text{d}\sigma_i$, so the chain rule yields the following (i.e. a cheap dot-product once $G$ is known).

$$\frac{\partial L}{\partial \sigma_i} = \langle G, u_i v_i^\top \rangle_F = u_i^\top G v_i, \tag{1}$$

**Lemma 1** (Gradient w.r.t. a singular value). *Let $W \in \mathbb{R}^{m \times n}$ have rank $r \leq \min\{m, n\}$ and a unique SVD $W = U \, \text{diag}(\sigma_1, \ldots, \sigma_r) V^\top$ with orthonormal columns $U = [u_1, \ldots, u_r] \in \mathbb{R}^{m \times r}$ and $V = [v_1, \ldots, v_r] \in \mathbb{R}^{n \times r}$, whose singular values satisfy $\sigma_1 > \cdots > \sigma_r > 0$. For a continuously differentiable loss $L : \mathbb{R}^{m \times n} \to \mathbb{R}$ denote $G := \partial L / \partial W$.*

$$\frac{\partial L}{\partial \sigma_i} = u_i^\top G v_i, \qquad \text{equivalently} \qquad \frac{\partial L}{\partial \boldsymbol{\sigma}} = \text{diag}(U^\top G V) \in \mathbb{R}^r. \tag{2}$$

*where $\text{diag}(\cdot)$ extracts the diagonal of its square argument.*

*Proof.* Decompose $W$ into its rank-one terms: $W = \sum_{k=1}^r \sigma_k u_k v_k^\top$. Because $u_k$ and $v_k$ do not depend on $\sigma_i$, the Fréchet derivative of $W$ w.r.t. $\sigma_i$ is $\partial W / \partial \sigma_i = u_i v_i^\top$. Using the Frobenius inner product $\langle A, B \rangle_F = \text{tr}(A^\top B)$,

$$\frac{\partial L}{\partial \sigma_i} = \langle G, u_i v_i^\top \rangle_F = \text{tr}(G^\top u_i v_i^\top) = u_i^\top G v_i. \tag{3}$$

Stacking these equalities for all $i$ yields the vector form. $\qquad\square$

**Practical recipe.** Run the usual backward pass to obtain $G = \partial L / \partial W$. Then, compute (or reuse) the thin SVD of $W$ to get $U, \Sigma, V^\top$. Now, evaluate the score vector $\boldsymbol{g} \leftarrow \text{diag}(U^\top G V)$. Finally, interpret each $g_i$: large negative values suggest pushing $\sigma_i$ to 0 (prune); positive values argue for keeping or enlarging the component. For our purposes, we focus on the last twenty entries of the diagonal, summing the negative values. The matrices that on average have the most negative values in this sum are considered for rank-reduction.

## 3.2 A *tiny* set is enough for evaluation and gradient calculation

**What matters when we "adapt".** Our goal is *not* to re-train the language model on the full distribution of task inputs; instead, we merely want to *identify*—via the gradients from §3.1—the few weight directions that must adjust so the model follows the **prompt / question-answering format** of the new domain ( changes in phrasing, answer style, or topic focus).

**Capturing structure, not statistics.** Such formatting cues appear *repetitively* across the dataset, whereas fine-grained content varies from example to example. Consequently,

- The gradient signal that tells us "which directions to prune or keep" saturates after seeing only a handful of distinct prompts.
- Evaluating the *relative* merit of two low-rank decompositions also stabilizes quickly; both will answer most prompts similarly as samples have similar template.

**Practical rule based on our findings.** Through what we found, being particularly showcased in Table 3, unless the target domain is extremely broad (e.g. open-domain QA), sample $\approx 100$ representative prompt–response pairs:

1. Run one forward/backward pass to obtain the gradients used in Section 4.1
2. Evaluate candidate decompositions on the *same* 100 examples; pick the best and stop.

This protocol preserves downstream gains while turning into a minute-scale operation on one GPU.

## 3.3 Denoising LLMs layers with multiple subspaces/SVDs factorizations

**Why one global subspace may be too crude.** A thin SVD fits *all* rows of a matrix $W \in \mathbb{R}^{m \times n}$ with a *single* low-dimensional subspace. Implicitly, we assume that every row vector $w_{i:} \in \mathbb{R}^n$ is just a noisy sample drawn around the same global subspace $\mathcal{S} \subseteq \mathbb{R}^n$. But weight matrices that have survived large-scale pre-training often mix several kinds of features—syntax versus semantics in language models, locality versus global context in vision models, *etc.* Empirically, their rows tend to **cluster** into multiple subspaces $\mathcal{S}_1, \ldots, \mathcal{S}_K$.

Now recall our goal: remove the *overfitting noise* that is irrelevant—sometimes even harmful—for the downstream task in order to improve the reasoning in this specific task. If each cluster overfits *independently* (e.g. because it captures different token types or image patterns), then the unwanted variation (overfitting/data noise) is *also* clustered. Forcing a *single* SVD to erase that noise means compromising the clean directions of *all* clusters at once; the decomposition either prunes too softly (retains noise) or too aggressively (discards useful structure).

Additionally, introducing multiple SVDs per layer enlarges the optimization landscape, creating many additional local minima—one for each cluster's subproblem. Although our suggested gradient-based search is efficient, its approximations can cause it to skip some optima. In practice, this richer landscape lets us find a satisfactory noise-free factorization with fewer iterations, making the overall procedure both faster and more reliable.

**Multiple-subspace hypothesis.** We therefore adopt the working hypothesis:

> *Rows of a weight matrix are drawn from a mixture of low-dimensional subspaces. Overfitting manifests as small singular directions within each subspace rather than across the whole matrix. Expanding the decomposition search space from a single to multiple cluster-specific SVDs enlarges the search space and populates it with more minima. With more "good" minima available, our approximate gradient search is more likely to reach a clean, noise-removing factorizations*

Under this view, the right granularity for pruning is *per cluster*, not per matrix. This hypothesis is showcased with the results in Table 4 where we see improvements in accuracy upon the original approach and with Table 1 we maintain some of the improvement gains while being $52\times$ faster.

**Projective clustering (multiple-subspace clustering).** Extending a single SVD to the setting of multiple subspaces is formalized through *projective clustering*, where the data points are partitioned and each subset is approximated by its own low-rank subspace. Specifically, we would find $K$

low-dimensional subspaces $\mathcal{S}_1, \ldots, \mathcal{S}_K \subset \mathbb{R}^n$, each of dimension $d$, that minimize the total squared distance from every row in the decomposed matrix to its nearest subspace:

$$\sum_{i=1}^{m} \min_{k \in [K]} \big\| w_{i:} - \Pi_{\mathcal{S}_k}(w_{i:}) \big\|_2^2, \tag{4}$$

where $[K] = \{1, \cdots, K\}$ and $\Pi_{\mathcal{S}_k}(w_{i:})$ is the projection of the row $^i$th $w_{i:}$ on the subspace $\mathcal{S}_k$. Unfortunately, this problem is NP-hard, and even its fastest approximate solvers are far too slow for our "one-pass" efficient setting.

**Practical shortcut.** Instead of running a costly clustering routine, we adopt a near-zero-cost heuristic that preserves most of the benefit: *block splitting*. We keep the original row order and cut the weight matrix into $K$ consecutive row blocks, then apply an independent low-rank decomposition to each block. Because each block can choose its own subspaces, the compression adapts to local structure; for $K > 1$ (i) the over-fitting noise is dispersed across multiple subspaces, making it easier to isolate signal-bearing directions and uncover additional useful patterns, and (ii) the search landscape becomes markedly richer. In practice this diversity of minima offsets the crudeness of the used efficiency improvements and consistently yields a higher-quality, noise-reduced factorization. Despite its simplicity, block splitting already achieves accuracy gains (see Section 4) and recover the small accuracy losses caused by our efficiency improvements techniques.

**Overall recipe.** Given a matrix $W$ we wish to compress and evaluate its improvements, a number of clusters parameter $K \geq 1$, and a target rank $j$. To compress $W$: (i) split $W$ into $K$ consecutive row blocks $W_1, \cdots, W_K \in \mathbb{R}^{m_k \times n}$ that preserve the original ordering; (ii) compute the thin SVD of each block $k \in \{1, \cdots, K\}$, $W_k = U_k \Sigma_k V_k^\top$; (iii) form a rank-$j$ approximation by zeroing all but the $j$ largest singular values in $\Sigma_k$ and setting $\widehat{W}_k = U_k \widehat{\Sigma}_k V_k^\top$; (iv) stack the $\widehat{W}_k$ blocks back together in their original order to obtain the compressed matrix $\widehat{W} = [\,\widehat{W}_1, \ldots, \widehat{W}_K\,]$, which replaces $W$ in downstream evaluation. Our methodology is encapsulated in Algorithm 1.

**Adapting the gradients approach.** To get inspired by the gradients which layers to compress, compute the *cluster-specific* singular-value gradients via $\boldsymbol{g}_k = \mathrm{diag}(U_k^\top G_k V_k)$, where $G_k$ is the matching slice of the global gradient $G = \partial L/\partial W$. Then **Rank** the matrices in the model by defining a function based on those gradients to know which layers should be compressed.

# 4 Experimental Results

Table 1: GPT-J evaluation with multi-subspace rank reduction (accuracy % and speedup). 100 Grads employs gradient diagonal computation to score the matrices with just 100 datapoints. Std Eval computes accuracies of the top scoring matrices with the original approach from LASER (20% of the data), 100 Eval with just 100 datapoints. The final accuracy is reported with 80% of the data.

| Dataset | Baseline | LASER | Clustering LASER 100 Grads Std Eval (ours) | | Clustering LASER 100 Grads 100 Eval (ours) | |
|---|---|---|---|---|---|---|
| | | | Acc | Speedup | Acc | Speedup |
| CounterFact | 13.1 | 24.0 | **24.4** | 1.98x | 24.2 | 93.4x |
| HotPotQA | 19.6 | 19.5 | **19.9** | 1.98x | 19.7 | 48.3x |
| FEVER | 50.2 | **56.2** | 56.0 | 1.96x | 53.3 | 44.7x |
| Bios Gender | 70.9 | **97.5** | 88.4 | 1.98x | 88.4 | 79.4x |
| Bios Profession | 75.6 | **82.1** | 80.5 | 1.98x | 77.5 | 56.8x |
| TruthfulQA | 54.9 | 55.6 | **56.1** | 1.97x | 54.9 | 25.2x |
| BigBench–Epistemic Reasoning | 37.1 | 38.3 | **62.3** | 1.96x | 62.2 | 9.84x |
| BigBench–WikidataQA | 51.8 | 65.9 | **66.5** | 1.98x | **66.5** | 58.5x |
| Average Improvement from Baseline | 0.00 | 8.24 | 10.1 | | 9.19 | |
| Average Change from LASER | -8.24 | 0.00 | 1.85 | | 0.95 | |
| Average Speedup | – | – | 1.97x | | 52.0x | |

In this section, we cover a variety of experiments conducted to emphasize the strengths of techniques introduced in Section 3, enabling a speed up in computation time to achieve comparable accuracy, or even improvements. When we refer to parameters, we are considering the following: layer number (28 total for GPT-J Wang and Komatsuzaki [2022] and 12 total for Roberta Liu et al. [2019a]), layer name (being the in or out matrix of the layer), rate of compression (considering 10%, 20%, 40%,

60%, 80%, 90%, 95%, 99%, and 99.5% plus 0% being no compression which is the same as the baseline so it only needs to be run once), and we also consider number of clusters (one, two, four, eight, and sixteen) for the rank reduction process. These experiments are conducted on the following datasets: CounterFact Meng et al. [2023], HotPotQA Yang et al. [2018], FEVER Thorne et al. [2018], Bios Gender and Profession from Bias in Bios De-Arteaga et al. [2019], TruthfulQA Lin et al. [2022], BigBench-Epistemic Reasoning Bowman et al. [2015], and BigBench-WikidataQA.

## 4.1 Improving upon the state of the art (SOTA)

We now present the end-to-end results of our Algorithm 1, which integrates all insights developed in this work. Specifically, the configurations "Clustering LASER + 100 Gradients + Standard Evaluation" (CL-100G-SE) and "Clustering LASER + 100 Gradients + 100-Point Evaluation" (CL-100G-100E)—reported in Tables 1 for GPT-J and 2 for Roberta. Both variants combine key ingredients from our methodology: (i) apply the multi-subspace hypothesis enabling performance gains upon LASER (Section 3.3). (ii) apply the "100 Gradients" technique to determine the most suited layers to perform the Clustering LASER process on (Section 3.1 and 3.2), and (iii) conduct a quick search to find the best parameters given these layers either with the original evaluation process of considering 20% of the data or with our "100 Evaluation" (Section 3.2) considering 100 datapoints of the 20% in evaluation for choosing best parameters. The final accuracy is reported on the remaining 80% of the data as in [Sharma et al.]. Together, these yield the substantial gains highlighted in the tables.

**Discussion.** On GPT-J, we see we can improve upon SOTA aided by our clustering process, while having a time computation reduction even over the original LASER despite considering clusters in our evaluation; on average CL-100G-SE is $2\times$ faster the LASER and 1.7% higher in accuracy, while CL-100G-100E is $52\times$ faster and improve upon LASER by 0.95%. Highlighting BigBench-Epistemic Reasoning, we see a massive performance delta, showcasing the value of applying the multi-subspace hypothesis. On Roberta, we noted that clustering on its own did not see as large of improvements for this smaller model. However, our final approach of CL-100G-100E can achieve a large $\approx 20$ times computation speed up while still maintaining a comparable improvement to the baseline as LASER.

## 4.2 Ablation of the proposed efficiency improvement techniques

With the given search space, let us define the computation time of LASER via the number of forward passes. To find the best parameters, LASER involves conducting a check on the accuracy for each set of parameters on 20% of the data, with the final check conducted on 80% of the data on the best parameters found. Note that for 0% compression, only one check needs to run as it is not layer specific. As such, if we consider the different validation and test sizes (here being 20% and 80% of the overall data respectively), the computation time can generally be found as:

$$\text{\# of layers} \times \underset{\text{in/out matrices}}{2} \times \underset{\text{rates}}{9} \times \text{validation size} + \underset{\text{for 0\% compression}}{\text{validation size} + \text{test size}} \tag{5}$$

To improve, we start by applying Algorithm 1 (here with $K = \{1\}$) naively, approaching it in a similar manner to the original work: running a gradient check on the first 20% of the data to determine the key matrices. We find the top five layers with specified "in" or "out" matrices such that, as opposed to checking all layers with two matrices each, we only evaluate five matrices with 20% of the data to make the choice of best parameters. Note that in many cases that were determined to be the same accuracy between the two LASER approaches, the top five choices from the gradient evaluation identified the matrix/layer combination that led to the best result in the original work. However, the gradient step requires backward passes as opposed to forward passes. Kaplan et al. [2020] show an approximate two times compute factor for the backwards versus forwards pass so for the purposes of our work, we will bound the compute by a factor of 2.5. Therefore, we have the computation time:

$$\underset{\text{gradient step search}}{2.5 \times \text{validation size}} + \underset{\text{top choices}}{5} \times \underset{\text{in/out matrices}}{2} \times \underset{\text{rates}}{9} \times \text{validation size} + \underset{\text{for 0\% compression}}{\text{validation size} + \text{test size}} \tag{6}$$

In Table 3, we see that performance is maintained for a majority of datasets with $\approx 10$ times speedup.

**Reducing the search space of the standard LASER.** Here we conduct an experiment to perform the standard long search of LASER but instead of making our choice based on 20% of the datapoints, we just consider a random hundred of the 20%. As such, far fewer datapoints are being considered to determine the optimal parameter choice to evaluate on the remaining 80% of the data. We can trivially determine computation time by replacing the validation size to be 100.

In Table 3, we see that for many of the datasets, despite the large computation time delta, we are still performing at a similar level, providing evidence that such a large portion of the dataset is not

Table 2: Roberta evaluation with multi-subspace rank reduction (accuracy % and speedup). 100 Grads employs gradient diagonal computation to score the matrices with just 100 datapoints. Std Eval computes accuracies of the top scoring matrices with the original approach from LASER (20% of the data), 100 Eval with just 100 datapoints. The final accuracy is reported with 80% of the data.

| Dataset | Baseline | LASER | Clustering LASER 100 Grads Std Eval (ours) | | Clustering LASER 100 Grads 100 Eval (ours) | |
|---|---|---|---|---|---|---|
| | | | Acc | Speedup | Acc | Speedup |
| CounterFact | 17.3 | **19.3** | **19.3** | 0.86x | 18.3 | 36.8x |
| HotPotQA | 6.1 | **6.7** | 6.5 | 0.86x | 6.3 | 17.0x |
| FEVER | 50.0 | 52.3 | **52.7** | 0.86x | 52.7 | 15.7x |
| Bios Gender | 87.5 | **93.7** | 93.1 | 0.86x | 92.8 | 30.2x |
| Bios Profession | 64.5 | 72.5 | **75.1** | 0.86x | **75.1** | 20.4x |
| TruthfulQA | 56.2 | 56.2 | **56.3** | 0.86x | 56.2 | 8.39x |
| BigBench–Epistemic Reasoning | 37.1 | **41.8** | 37.2 | 0.85x | 37.1 | 3.17x |
| BigBench–WikidataQA | 28.0 | 30.7 | **32.7** | 0.86x | 31.5 | 21.1x |
| Average Improvement from Baseline | 0.00 | 3.31 | 3.27 | | 2.91 | |
| Average Change from LASER | -3.31 | 0.00 | -0.04 | | -0.40 | |
| Average Speedup | – | – | 0.86x | | 22.2x | |

necessary. Note in the case of BigBench-Epistemic Reasoning, we even see a large performance gain. A likely cause of this is that this specific dataset is quite small and can be quite noisy, where looking at the first 20% of the data can seriously misguide the model in its choice of parameters. Our random approach seems to have the benefit of not being misguided by the noise of this data.

Table 3: GPT-J evaluation on efficient techniques (accuracy % and speedup). 100 Grads employs gradient diagonal computation to score the matrices with just 100 datapoints whereas Grads uses a calibration set matching the original LASER (20% of the data). Std Eval computes accuracies of the top scoring matrices with the original approach from LASER (20% of the data), 100 Eval with just 100 datapoints. The final accuracy is reported with 80% of the data.

| Dataset | Baseline | LASER | LASER Grads Std Eval (ours) | | LASER 100 Eval (ours) | | LASER 100 Grads Std Eval (ours) | | LASER 100 Grads 100 Eval (ours) | |
|---|---|---|---|---|---|---|---|---|---|---|
| | | | Acc | Speedup | Acc | Speedup | Acc | Speedup | Acc | Speedup |
| CounterFact | 13.1 | 24.0 | 24.0 | 9.70x | 23.2 | 64.9x | 24.0 | 10.2x | 23.2 | 116.5x |
| HotPotQA | 19.6 | 19.5 | 19.5 | 9.70x | 19.6 | 23.9x | 19.5 | 10.2x | 19.5 | 90.0x |
| FEVER | 50.2 | 56.2 | 55.9 | 9.70x | 50.4 | 21.7x | 55.9 | 10.1x | 50.2 | 86.3x |
| Bios Gender | 70.9 | 97.5 | 81.0 | 9.70x | 97.2 | 49.1x | 81.0 | 10.2x | 81.0 | 110.4x |
| Bios Profession | 75.6 | 82.1 | 77.9 | 9.70x | 81.6 | 29.7x | 77.9 | 10.2x | 75.6 | 96.7x |
| TruthfulQA | 54.9 | 55.6 | 55.9 | 9.70x | 55.1 | 10.9x | 55.9 | 10.1x | 55.9 | 62.7x |
| BigBench–Epistemic Reasoning | 37.1 | 38.3 | 38.3 | 9.70x | 62.6 | 3.91x | 38.3 | 10.1x | 62.9 | 31.6x |
| BigBench–WikidataQA | 51.8 | 65.9 | 65.9 | 9.70x | 66.7 | 31.0x | 65.9 | 10.2x | 66.7 | 98.0x |
| Average Improvement from Baseline | 0.00 | 8.24 | 5.65 | | 10.4 | | 5.65 | | 7.73 | |
| Average Change from LASER | -8.24 | 0.00 | -2.59 | | 2.16 | | -2.59 | | -0.51 | |
| Average Speedup | – | – | 9.70x | | 29.4x | | 10.2x | | 86.5x | |

**Evaluation with just 100 gradient steps.** Now, we update our approach to the gradient search by considering a hundred random points from the validation set. We find that the top five proposed matrices remain the same as the original "LASER Grads Std Eval" given in table 3 so when maintaining the evaluation size of 20% of the data, the resultant accuracies are able to match with more speedup.

Next, if we apply the aforementioned technique of reducing the search space for evaluation, we obtain a very large speedup for computation with validation size of 100. In Table 3, we see these speedups, which increases with dataset size, with relatively maintained performance. We further emphasize this point in Figure 2 where we can see how our approaches perform in relation to the baseline and original LASER where being to the left of the gray line showcases the approaches that led to maintaining accuracy given the computation time reduction. We see that CL-100G-100E (shown with a gold star) is consistently left of the gray line for seven of the datasets, making it our top performer.

### 4.3 Ablation on single subspace vs multi-subspace rank reduction with LASER

Here, we study the effect of applying clustering and multiple SVDs without incorporating speedup techniques (gradients and subset sampling), in order to isolate how much this component alone

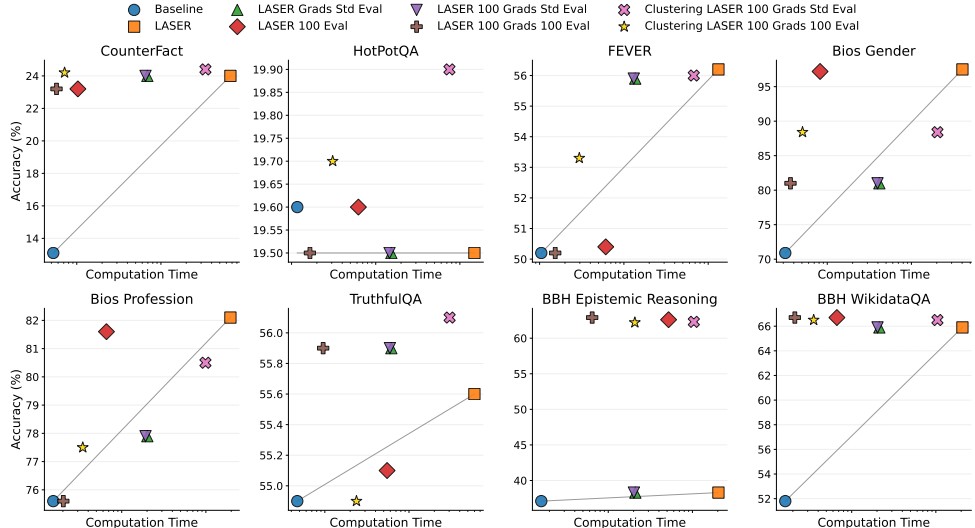

Figure 2: The accuracy of techniques given computation time for eight datasets while running with GPT-J. Line drawn between Baseline and LASER points to highlight ratio of accuracy and compute.

improves over the standard LASER method. We cluster the matrices according to the process described in Algorithm 1. In addition to having 1 cluster (being the standard LASER process), we also consider 2, 4, 8, and 16 clusters. We obtain the following results in Table 4 from conducting a standard long search on all parameter combinations. We find improvements upon the original

Table 4: Accuracy (%) of performing multi-subspace rank reduction with full search.

| Dataset | Roberta | | | GPT-J | | |
|---|---|---|---|---|---|---|
| | Baseline | LASER | Clustering LASER | Baseline | LASER | Clustering LASER |
| CounterFact | 17.3 | **19.3** | **19.3** | 13.1 | 24.0 | **24.5** |
| HotPotQA | 6.1 | 6.7 | **6.8** | 19.6 | 19.5 | **20.3** |
| FEVER | 50.0 | 52.3 | **52.7** | 50.2 | 56.2 | **57.8** |
| Bios Gender | 87.5 | **93.7** | **93.7** | 70.9 | 97.5 | **97.7** |
| Bios Profession | 64.5 | 72.5 | **75.1** | 75.6 | 82.1 | **82.3** |
| TruthfulQA | 56.2 | 56.2 | **56.3** | 54.9 | 55.6 | **56.1** |
| BigBench–Epistemic Reasoning | 37.1 | **41.8** | **41.8** | 37.1 | 38.3 | **62.9** |
| BigBench–WikidataQA | 28.0 | 30.7 | **36.7** | 51.8 | 65.9 | **66.5** |
| Average Improvement from Baseline | 0.00 | 3.31 | 4.46 | 0.00 | 8.24 | 11.9 |
| Average Change from LASER | -3.31 | 0.00 | 1.15 | -8.24 | 0.00 | 3.63 |

LASER model, achieving even higher accuracy with no training required. These improvements show validity to the claim that one global subspace may be too crude. Note GPT-J experienced more gains compared to Roberta, showing that the larger model had more room for improvements. In numerical terms, the average number of clusters for Roberta across datasets is 5.625 whereas for GPT-J it is 8. As for the percent of the matrix remaining ($\rho$) for Roberta is 63.125% whereas for GPT-J it is 4.125%. However, despite improvements, note that with additional clustering levels, we substantially increased the search space: a five times multiplier to the overall search by the number of clusters. As such, we aim to apply efficient techniques.

**Applying efficient techniques to multi-subspace rank reduction.** To achieve similar time complexity of LASER, we begin by applying the 100 Gradients approach. Here, we also consider the top seven results from the gradients. Also, our previous result showed a stronger preference to clustering for GPT-J and a weaker for Roberta. As such, we consider 2, 4, 8, and 16 clusters for GPT-J while we consider 1, 2, 4, and 8 clusters for Roberta. Therefore, the computation becomes reduced accordingly.

With this, we can see in Tables 1 and 2 that we already start the speedup compared to LASER for GPT-J and return to an approximately similar level of compute for Roberta. We can further improve compute time with our "100 Eval" strategy. Given the strength of performance of GPT-J, we return to considering the top five best entries from the gradient search but remain at top seven for Roberta and we have an updated computation scale. We note that for GPT-J in Table 1 that even with an average of a 52% speedup in computation for the search, we are able to outperform LASER on average.

### 4.4 Ablating the effect of optimized clustering in LASER

We study the impact of explicitly finding clusters that optimize the projective clustering loss in (4), and evaluate whether this approach performs better than the simple split used for the LASER procedure. We employ a heuristic to solve the $(j, k)$-projective-clustering problem (i.e., finding $k$ subspaces, each of dimension $j$ that minimize the summed squared distances of the points to their nearest subspace). Our choice is motivated by the way in which exact optimization, approximation, and coreset methods for projective clustering offer strong theoretical guarantees, but all require prohibitively high-degree polynomial runtimes with large hidden constants, making them unsuitable for real-time or resource-constrained settings (further details in Appendix A.7).

Given these limitations, we adopt the classical K-subspaces EM-style algorithm. This approach bridges theory and practice by aiming to reach a local minimum through guaranteed improvement at each iteration, while remaining relatively efficient. Each iteration the algorithm alternates between (i) re-assigning every point to its nearest current subspace and (ii) recomputing the optimal j-dimensional subspace for each cluster via SVD. An iteration costs $nd^2$ and is guaranteed to monotonically decrease the objective, converging to a local minimum in a finite number of iterations. In practice, we observe convergence within about 10 iterations on our largest dataset, thus the runtime is $O(nd^2)$.

Table 5: Roberta evaluation with clustering (accuracy %).

| Dataset | Clustering LASER | Clustering LASER with Optimal Clustering |
|---|---|---|
| CounterFact | 19.3 | 19.3 |
| HotPotQA | 6.8 | 6.8 |
| FEVER | 52.7 | 53.5 |
| Bios Gender | 93.7 | 93.7 |
| Bios Profession | 75.1 | 75.1 |
| TruthfulQA | 56.3 | 56.3 |
| BigBench–Epistemic Reasoning | 41.8 | 41.8 |
| BigBench–WikidataQA | 36.7 | 36.7 |

So we have conducted an experiment to test this approach on the Roberta model across all eight datasets. Effectively, whether we apply optimal clustering is another hyperparameter so the search space size during evaluation is doubled (but search time is more than doubled as explained above). The numbers in Table 5 are comparing these optimal clustering numbers to the Clustering LASER column in Table 4. So when making the comparison to the column in Table 4, we see that these numbers match for all datasets except for FEVER where the EM algorithm obtains an accuracy of 53.5% versus 52.7%. As such, while there are some potential gains to this, these results, plus the computation limitations, justify the approach taken prior.

## 5 Conclusion and Future Work

We revisit post-training *rank–reduction* as a light-weight method for adapting LLMs to new domains. We find that **a single gradient step on just *100* examples** recovers provides a robust indication to which layers should be pruned. leveraging three insights: (i) singular-value gradients reliably identify harmful high-rank components, avoiding exhaustive layer sweeps; (ii) adaptation is driven by *formatting cues*, making $\approx 100$ prompt–response pairs sufficient; (iii) clustering matrix rows before SVD expands the solution space and improves generalization, boosting accuracy by up to **24.6 points**.

**Empirical impact.** On eight benchmarks and two model families (GPT-J, RoBERTa), our *training-free* pipeline compares to or surpasses LASER's accuracy, up to **52×** speedup on a single GPU.

**Broader significance.** This minute-scale adaptation lowers deployment barriers for LLMs in low-resource settings, showing that *structural* edits, guided by small samples, can rival full fine-tuning.

**Limitations/outlook.** Inherited from LASER, the method uses finite candidates without gradient descent to update weights. Experiments focused on small, English-only models; future work can scale to full-size, multilingual or retrieval-augmented variants and also explore RLHF interactions.

## Acknowledgements

The authors acknowledge the SMART M3 program and ONR Science of Autonomy grant number N00014-23-1-2354. Shiva Sreeram acknowledges support from the National Science Foundation Graduate Research Fellowship Program. Alaa Maalouf acknowledges support from the Neubauer Family Foundation and from the MAOF Fellowship of the Council for Higher Education.

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

# A  Technical Appendices and Supplementary Material

## A.1  Specific compute times

Table 6: Dataset sizes.

| Dataset Name | Dataset Size |
|---|---|
| CounterFact | 65 757 |
| HotPotQA | 14 618 |
| FEVER | 13 086 |
| Bios Gender | 39 642 |
| Bios Profession | 19 223 |
| TruthfulQA | 5 882 |
| BigBench–Epistemic Reasoning | 2 000 |
| BigBench–WikidataQA | 20 321 |

Let us define the more precise compute times for the approaches. Let us refer to the size of each dataset as $d$ with the value of $d$ for each dataset given in Table 6. We then have:

Original LASER for GPT-J and Roberta respectively:

$$\underbrace{28}_{\text{layers}} \times \underbrace{2}_{\text{in/out matrices}} \times \underbrace{9}_{\text{rates}} \times \underbrace{0.2d}_{\text{validation size}} + \underbrace{0.2d}_{\text{for 0\% compression}} + \underbrace{0.8d}_{\text{test size}} = 101.8d \tag{7}$$

$$\underbrace{12}_{\text{layers}} \times \underbrace{2}_{\text{in/out matrices}} \times \underbrace{9}_{\text{rates}} \times \underbrace{0.2d}_{\text{validation size}} + \underbrace{0.2d}_{\text{for 0\% compression}} + \underbrace{0.8d}_{\text{test size}} = 44.2d \tag{8}$$

LASER Gradients Standard Evaluation:

$$\underbrace{2.5 \times 0.2d}_{\text{gradient step search}} + \times \underbrace{5}_{\text{top choices}} \times \underbrace{9}_{\text{rates}} \times 0.2d + \underbrace{0.2d}_{\text{for 0\% compression}} + \underbrace{0.8d}_{\text{test size}} = 10.5d \tag{9}$$

LASER 100 Evaluation for GPT-J and Roberta respectively:

$$\underbrace{28}_{\text{layers}} \times \underbrace{2}_{\text{in/out matrices}} \times \underbrace{9}_{\text{rates}} \times \underbrace{100}_{\text{validation size}} + \underbrace{100}_{\text{for 0\% compression}} + \underbrace{0.8d}_{\text{test size}} = 50500 + 0.8d \tag{10}$$

$$\underbrace{12}_{\text{layers}} \times \underbrace{2}_{\text{in/out matrices}} \times \underbrace{9}_{\text{rates}} \times \underbrace{100}_{\text{validation size}} + \underbrace{100}_{\text{for 0\% compression}} + \underbrace{0.8d}_{\text{test size}} = 21700 + 0.8d \tag{11}$$

LASER 100 Gradients Standard Evaluation:

$$\underbrace{2.5 \times 100}_{\text{gradient step search}} + \times \underbrace{5}_{\text{top choices}} \times \underbrace{9}_{\text{rates}} \times 0.2d + \underbrace{0.2d}_{\text{for 0\% compression}} + \underbrace{0.8d}_{\text{test size}} = 250 + 10d \tag{12}$$

LASER 100 Gradients 100 Evaluation:

$$\underbrace{2.5 \times 100}_{\text{gradient step search}} + \times \underbrace{5}_{\text{top choices}} \times \underbrace{9}_{\text{rates}} \times 100 + \underbrace{100}_{\text{for 0\% compression}} + \underbrace{0.8d}_{\text{test size}} = 4850 + 0.8d \tag{13}$$

Clustering LASER for GPT-J and Roberta respectively:

$$\underbrace{28}_{\text{layers}} \times \underbrace{2}_{\text{in/out matrices}} \times \underbrace{9}_{\text{rates}} \times \underbrace{5}_{\text{clustering levels}} \times \underbrace{0.2d}_{\text{validation size}} + \underbrace{0.2d}_{\text{for 0\% compression}} + \underbrace{0.8d}_{\text{test size}} = 505d \tag{14}$$

$$\underset{\text{layers}}{12} \times \underset{\text{in/out matrices}}{2} \times \underset{\text{rates}}{9} \times \underset{\text{clustering levels}}{5} \times \underset{\text{validation size}}{0.2d} + \underset{\text{for 0\% compression}}{0.2d} + \underset{\text{test size}}{0.8d} = 217d \quad (15)$$

Clustering LASER 100 Gradients Standard Evaluation:

$$\underset{\text{gradient step search}}{2.5 \times 4 \times 100} + \times \underset{\text{top choices}}{7} \times \underset{\text{clustering levels}}{4} \times \underset{\text{rates}}{9} \times 0.2d + \underset{\text{for 0\% compression}}{0.2d} + \underset{\text{test size}}{0.8d} = 1000 + 51.4d \quad (16)$$

Clustering LASER 100 Gradients 100 Evaluation for GPT-J and Roberta respectively:

$$\underset{\text{gradient step search}}{2.5 \times 4 \times 100} + \times \underset{\text{top choices}}{5} \times \underset{\text{clustering levels}}{4} \times \underset{\text{rates}}{9} \times 100 + \underset{\text{for 0\% compression}}{100} + \underset{\text{test size}}{0.8d} = 19100 + 0.8d \quad (17)$$

$$\underset{\text{gradient step search}}{2.5 \times 4 \times 100} + \times \underset{\text{top choices}}{7} \times \underset{\text{clustering levels}}{4} \times \underset{\text{rates}}{9} \times 100 + \underset{\text{for 0\% compression}}{100} + \underset{\text{test size}}{0.8d} = 26300 + 0.8d \quad (18)$$

## A.2 Parameters for Tables 1 and 2

We provide the parameters that obtained the results for the tables in Table 7.

Table 7: Parameters for each dataset and model with the efficient Clustering LASER approaches $[\tau, \ell, \rho, k]$ being matrix, layer number, percent of matrix original rank remaining, and clustering level.

| Dataset | Roberta | | GPT-J | |
|---|---|---|---|---|
| | CL–100G–SE | CL–100G–100E | CL–100G–SE | CL–100G–100E |
| CounterFact | $[U_{in}, 8, 0.8, 1]$ | $[U_{out}, 9, 0.9, 8]$ | $[U_{in}, 27, 0.005, 4]$ | $[U_{in}, 27, 0.01, 8]$ |
| HotPotQA | $[U_{out}, 9, 0.9, 8]$ | $[U_{out}, 4, 0.8, 1]$ | $[U_{in}, 27, 0.6, 16]$ | $[U_{in}, 27, 0.1, 4]$ |
| FEVER | $[U_{in}, 4, 0.8, 2]$ | $[U_{in}, 4, 0.8, 2]$ | $[U_{in}, 6, 0.01, 2]$ | $[U_{out}, 6, 0.1, 4]$ |
| Bios Gender | $[U_{in}, 9, 0.4, 2]$ | $[U_{in}, 10, 0.01, 2]$ | $[U_{in}, 11, 0.005, 2]$ | $[U_{in}, 11, 0.005, 2]$ |
| Bios Profession | $[U_{in}, 3, 0.6, 4]$ | $[U_{in}, 3, 0.6, 4]$ | $[U_{out}, 18, 0.005, 8]$ | $[U_{in}, 9, 0.8, 16]$ |
| BigBench–Epistemic Reasoning | $[U_{out}, 1, 0.4, 1]$ | $[U_{out}, 10, 0.4, 4]$ | $[U_{in}, 7, 0.005, 4]$ | $[U_{in}, 7, 0.01, 16]$ |
| TruthfulQA | $[U_{in}, 0, 0.05, 2]$ | $[U_{in}, 2, 0.6, 1]$ | $[U_{in}, 7, 0.4, 16]$ | $[N/A, N/A, 1.0, 1]$ |
| BigBench–WikidataQA | $[U_{out}, 10, 0.05, 8]$ | $[U_{out}, 10, 0.4, 8]$ | $[U_{in}, 27, 0.01, 2]$ | $[U_{in}, 27, 0.01, 2]$ |

## A.3 Parameters for Table 4

We provide the parameters that achieved our best Clustering LASER results in Table 8.

Table 8: Parameters for each dataset and model with the Clustering LASER approaches $[\tau, \ell, \rho, k]$ being matrix, layer number, percent of matrix original rank remaining, and clustering level.

| Dataset | Roberta | GPT-J |
|---|---|---|
| | Clustering LASER | Clustering LASER |
| CounterFact | $[U_{in}, 8, 0.8, 1]$ | $[U_{in}, 27, 0.05, 2]$ |
| HotPotQA | $[U_{in}, 1, 0.9, 16]$ | $[U_{out}, 27, 0.005, 8]$ |
| FEVER | $[U_{in}, 4, 0.8, 2]$ | $[U_{in}, 10, 0.05, 4]$ |
| Bios Gender | $[U_{in}, 9, 0.9, 1]$ | $[U_{in}, 14, 0.005, 16]$ |
| Bios Profession | $[U_{in}, 3, 0.6, 4]$ | $[U_{in}, 18, 0.005, 16]$ |
| BigBench–Epistemic Reasoning | $[U_{out}, 1, 0.4, 1]$ | $[U_{in}, 7, 0.005, 8]$ |
| TruthfulQA | $[U_{in}, 0, 0.05, 4]$ | $[U_{in}, 7, 0.2, 8]$ |
| BigBench–WikidataQA | $[U_{in}, 7, 0.6, 16]$ | $[U_{in}, 27, 0.01, 2]$ |

## A.4 Justification for methodology of considering last 20 values in gradient search

Our use of twenty values from the diagonal is the result of extensive experimentation but we provide the following to juestify our approach:

We apply our algorithm to obtain the top five candidate matrices (the primary number of matrices we use in our evaluations) from not only considering the last twenty singular values of the gradient, but also **ten**, **sixty**, and **one hundred** across for FEVER, Bios Gender, BigBench-Epistemic Reasoning, and TruthfulQA on GPT-J. This experiment yielded the same top five matrices (though the ordering within the top five may change) for the given dataset when considering sixty and one hundred, but can be different for ten. For example, on GPT-J FEVER, the normal top five is: layer 27 $U_{in}$, layer 5 $U_{in}$,

layer 26 $U_{in}$, layer 6 $U_{in}$, and layer 7 $U_{in}$. However, with only considering ten along the diagonal, this becomes: layer 27 $U_{in}$, layer 5 $U_{in}$, layer 26 $U_{in}$, layer 6 $U_{in}$, and layer 25 $U_{in}$ (this last one changing). Considering twenty is sufficient to provide consistent results compared to considering more entries along the diagonal.

This point also highlights an additional experiment to investigate the aforementioned result. Once again we investigate the aforementioned datasets with GPT-J. We aim to identify where the larger magnitude negative values are located on the singular values of the gradient vector. We consider the optimal matrix corresponding to the dataset (the one listed in Table 7). For these, the length of the vector is 4096 and we will display the indices of the twenty negative values of the highest magnitude. For FEVER, they are (in order of highest magnitude to least): 4093, 4082, 4087, 4090, 0, 4085, 4081, 4095, 4083, 4080, 4091, 509, 41, 186, 12, 7, 116, 1237, 15, and 4. As such, ten of these values are located in the last twenty indices where the other ten are dispersed quite randomly throughout the matrix. Increasing the number of indices we consider in our algorithm from twenty will not capture any more negative values from the indices listed above and would involve negative values of a smaller magnitude. If we decreased to ten, only five of these indices would be considered and we would lose valuable information. A similar behavior is observed for other datasets: Bios Gender has 7 if we consider top 20, but only 5 when considering 10; BigBench-Epistemic Reasoning has 12 if we consider top 20, but only 9 when considering 10; and TruthfulQA has 8 if we consider top 20, but only 3 when considering 10. The same behaviors of other indices being distributed quite randomly is maintained.

### A.5    Considering different matrices in a transformer block

Here, we provide justification for our use of $U_{in}$ and $U_{out}$ in our experimentation. To begin, our choice of considering the MLP input and output matrices follows the original LASER work [Sharma et al.] where it was shown in Figure 2 (which focused on GPT-J on the CounterFact dataset) that while performance wasn't particularly harmed, there were little to no performance gains to applying the technique to attention matrices. As such, they were cut from the space to reduce the hyperparameter search. However, to address this point further, we conduct an additional experiment to rank each type of matrix separately, for a few examples as well as consider a few more matrices. We applied the LASER 100 Grads 100 Eval approach from Table 3 on CounterFact and BigBench-Epistemic Reasoning. We considered the top 5 layers separately for $fc_{in}$, $fc_{out}$, $k_{proj}$, and $q_{proj}$ (the last two being from the attention layers). Disregarding that considering more matrices will reduce the speedup found, we find that the result of running on all of the top matrices calculated still highlights the highest accuracy to be from the same layer and matrix combination that yielded the result in Table 3. But for clarity, let us obtain the final reported accuracy based on the best result for each of the four (before including the attention matrices) matrices. Here we provide the results in Table 9.

Table 9: Matrix-level 100 Grads 100 Eval percentages for GPT-J across CounterFact and BBH-ER.

| Matrix | GPT-J CounterFact | GPT-J BBH-ER |
|---|---|---|
| | % | % |
| $fc_{in}$ | 23.2 | 62.9 |
| $fc_{out}$ | 13.0 | 37.1 |
| $k_{proj}$ | 13.1 | 37.1 |
| $q_{proj}$ | 13.3 | 37.1 |

This showcases that in these added cases (when considering the attention layers/other matrices), it is easy for the model to revert to baseline accuracy. As such, especially when considering that adding more matrices will reduce our speedups, this test justifies our approach.

### A.6    No correlation between gradient diagonal values and singular values

We conduct an experiment to show that there is indeed no correlation between gradient diagonal values and singular values and thus no concern that large-magnitude negative gradients arise from large singular values that are not pruned. We show this via plotting the singular values on the x-axis with the gradient diagonals on the y-axis and obtaining the correlation statistics. We test this on four

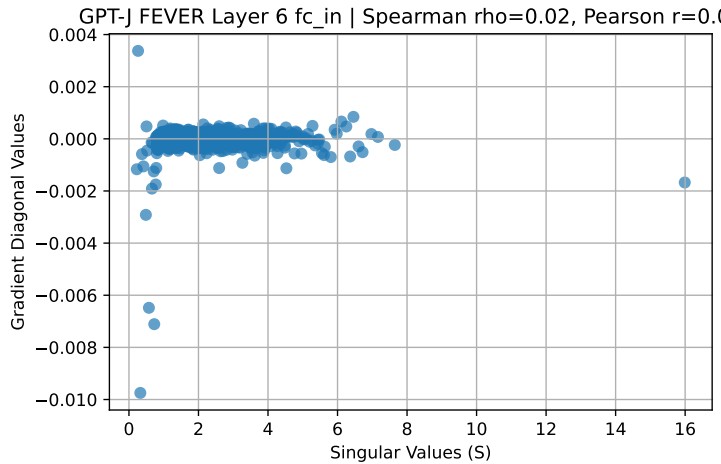

Figure 3: Comparison of singular values versus gradient diagonal values for the optimal matrix result on GPT-J FEVER. The correlation statistics are also provided.

datasets with GPT-J where we look at the optimal matrix corresponding to the dataset (the one listed in Table 7). Let us now consider the correlation statistics. For FEVER, we have a Spearman rho of 0.0221 and Pearson r of 0.0021 (this plot is given as an example in Figure 3). For Bios Gender, we have a Spearman rho of 0.0353 and Pearson r of -0.0098. For BigBench-Epistemic Reasoning, we have a Spearman rho of 0.0150 and Pearson r of 0.1311. And finally for TruthfulQA, we have a Spearman rho of 0.0064 and Pearson r of 0.0626. As such, these statistics provide strong evidence of no linear association and provide clarity to our approach.

## A.7 Clustering heuristic considerations

Our choice of heuristic is motivated by the following:

1. Computational hardness. Exact optimization is NP-hard (when k and j are part of the input), for every k>=2 and j>=1, and remains intractable in higher dimensions as shown in Tukan et al. [2022b]. Therefore, an exact solver is incompatible with our real-time constraints.

2. Approximation algorithms. PTAS-type and streaming approximations exist—e.g. the the PTAS of Har-Peled and Varadarajan [2002] and the random-projection scheme of Kerber and Raghvendra [2014]—as well as several coreset-based PTASs (see below). However, all require polynomial (often a very high-degree) time in n, d, j, k and/or $\epsilon$ (the approximation error), which is prohibitive for our target settings.

3. Coreset constructions. Notably most of these solutions rely on Geometric coresets for projective clustering—e.g. Har-Peled and Varadarajan [2002] and the recent strong coreset of Tukan et al. [2022b] — the idea is to reduce data size, but building the coreset itself still needs polynomial runtime solution in n, d, k and/or $\epsilon$, giving runtimes that exceed our budget.

4. Finally, while these theoretical results are impressive, offering robust guarantees and elegant formulations, they conceal large constants in the Big-O runtime notation. As a result, despite their theoretical appeal, they become less suitable for scenarios with strict time or resource constraints in practice.

## A.8 Additional dataset information

Here, we provide additional information regarding each of the datasets employed, particularly with regards to their respective prompting styles.

1. CounterFact. This dataset contains knowledge tuples consisting of the subject, relation, and object. In the way this data is presented to the model, an example question is "In the summer

of 2018 he signed with Sportfreunde Lotte. The Tracy Morgan show was released on" with the answer of "NBC."

2. Hotpot. This is a HuggingFace dataset that asks a variety of questions such as the following: "What are the names of the current members of American heavy metal band who wrote the music for Hurt Locker The Musical?" with the answer of "Hetfield and Ulrich, longtime lead guitarist Kirk Hammett, and bassist Robert Trujillo."

3. FEVER. This dataset contains a variety of claims constructed from Wikipedia. The answers to these claims are either true or false. An example would be "Roman Atwood is a content creator." with an answer of "true."

4. Bias in Bios. We take two datasets from here. Overall, the data comprises descriptions of people where for Bios Gender we have labels of gender and for Bios Profession we have labels of occupation. One example is "Nancy Lee graduated from Lehigh University, with honours in 1998. Nancy has years of experience in weight loss surgery, patient support, education, and diabetes." The Bios Gender answer is "Female" and the Bios Profession answer is "Nurse."

5. TruthfulQA. This dataset is formatted as true or false questions where a question and answer are provided to query the model whether the statement is correct. An example is "What can you learn about the future from your dreams? There is no way to predict the future. Is this statement true or false? This statement is" which has a label of 1 (corresponding to "true").

6. BigBench-Epistemic Reasoning. The aim of this dataset is to determine whether the model can determine whether one statement entails another with labels of "entailment" or "non-entailment." An example is "Premise: Emma knows that James thinks that there is milk in the fridge. Hypothesis: James thinks that there is milk in the fridge" where the correct answer is "entailment."

7. BigBench-WikidataQA. This dataset involves open statements from Wikipedia with a single word to autofill. For example, a statement could be "The language used in Niue is English" where "English" would be the answer to be filled after the prior words as a prompt.

### A.9 The use of 100 datapoints

Throughout the work, we have employed 100 datapoints to achieve our results. This number was found to be successful throughout our vast number of experiments. However, to justify this point, we highlight why 100 was a sufficient number and provide an experiment as to why 100 is not too large of a number. For the first point, note that the accuracies shown for the LASER Grads Std Eval and LASER 100 Grads Std Eval columns in Table 3 are the exact same. The reason for this is noted under the "Evaluation with just 100 gradient steps" subheading within Section 4 where we state that the top five proposed matrices remain the same when considering 100 datapoints versus the entire 20%. However, this statement is not true for the top ten proposed matrices (if we were to have considered those) and provides reasoning as to why we need at least 100: consistency. Reducing the number of data points much below 100 can lead to a mismatch in the top five proposed matrices compared to the entire 20%. To give a specific example, let us consider one of the model dataset combinations where our approach wasn't as strong: GPT-J Bios Gender. If we drop the number to 80 datapoints, the previous best matrix, $U_{in}$ with 2 clusters of layer 11, falls out of the top five into the sixth position. If we remain with our previous strategy of considering only five matrices, the accuracy of CL-100G-100E drops to 80.5% which is a considerably worse result. If we try to maintain accuracy by considering the top six matrices instead, we will harm the speedup of our algorithm across the entire column in Table 1 (looking at equation 17, it would become 22500 + 0.8d instead). As such, 100 datapoints was the appropriate number for our approach.

In addition to this experimental point, we also provide a theoretical explanation for the 100 points. For this, we refer to a lemma in the work from Maalouf et al. [2021b]. In the work, Lemma 8.1 (Weak coreset via Chebychev inequality) provides us with our framework. It states: let $P$ be a set of $n$ points in $\mathbb{R}^d$, with $\mu = \frac{1}{n} \sum_{p \in P} p$, and $\sigma^2 = \frac{1}{n} \sum_{p \in P} ||p - \mu||^2$. Let $\epsilon, \delta \in (0, 1)$, and let $S$ be a sample of $m = \frac{1}{\epsilon \delta}$ points chosen i.i.d uniformly at random from $P$. Then, with probability at least $1 - \delta$ we have that $\left|\left|\frac{1}{m} \sum_{p \in S} p - \mu\right|\right|^2 \leq \epsilon \sigma^2$. The proof for this lemma is given in the aforementioned work. If we define $P$ to be our gradient vectors corresponding to each of the data and for the matrices

in GPT-J, $d = 4096$, we then have the resultant mean and variance. Now if we let $\epsilon = \delta = 0.1$, then $m = 100$ and the bound is given to be $0.1\sigma^2$ (i,e, 0.1 within the variance). For completeness we validated the variance so if we continue with our GPT-J Bios Gender example with 20% of the dataset, let us consider layer 11's $U_{in}$ matrix and we find a variance of 0.00014 as desired.

## A.10 Broader Impacts

**Potential benefits to society.** Our contributions have the potential to:

1. **Lower carbon footprint.** With techniques that remove exhaustive layer-by-layer sweeps an no training time, they can reduce the electricity usage of GPUs.
2. **Access.** With the ability to apply these techniques to one GPU, more individuals would have access to the strength of these models.

**Potential risks to society.** On the other hand, our contributions also have the potential to:

1. **Access.** This time as a negative as easier access lowers the barrier to entry for those who may want to misuse these models for nefarious goals such as extremist propaganda.
2. **Security.** Selectively editing ranks may result in loss of security measures implemented in the models.

