# OpenReview forum: "Compress to Impress: Efficient LLM Adaptation Using a Single Gradient Step on 100 Samples"
_NeurIPS.cc/2025/Conference — NeurIPS 2025 spotlight_

### Official Review · Reviewer_UaEY · 2025-06-24

**Clarity:** 2
**Significance:** 3
**Originality:** 3
**Rating:** 4
**Confidence:** 3

**Summary:**

The paper builds on a pruning technique called LAyer SElective Rank reduction (LASER).
LASER improves the performance of a model on downstream task by replacing its weight matrices with their SVD approximations with reduced ranks.  LASER removes higher-rank components which correspond to smaller singular values. However, it requires a search for the reduction rate on the rank to prevent performance drop, which is not efficient.

This paper improves the efficiency of LASER by requiring less number of samples than LASER. Additionally, it provides a more reliable approximation than LASER by using multiple SVD projections per layer. These SVD projections are selected based on score vectors, which are calculated by replacing the sigma matrix in SVD with the gradients of loss with respect to the singular values.

The proposed method show comparable performance to LASER with certain speedups on GPT-J and RoBERTa. The tradeoff is different for different scenarios.

**Questions:**

You mention that the proposed method is training-free but this is not entirely correct as it still relies on gradients of the loss with respect to the singular values. This part may benefit from clarification.

100 seems like a magic number. Have you done any experiments on why 100 is a good choice? What happens for different number of samples? If you could show these results, that would add more support for the choice of 100.

I think writing needs to be improved too (see the details in weaknesses).

I would be happy to increase my scores (quality for the issues about the number of samples and clarity for the issues regarding the writing) if these changes are done. I would also increase my rating to Borderline accept.

**Ethical Concerns:**

["NO or VERY MINOR ethics concerns only"]

**Final Justification:**

The authors addressed my concerns. They provided evidence (both empirical and theoretical) for the motivation behind the number of samples used in their algorithm.

**Limitations:**

Yes

**Paper Formatting Concerns:**

No concerns

**Quality:**

2

**Strengths And Weaknesses:**

I can list some of the strengths of the proposed approach as follows:
- requires a small dataset for adapting models to downstream tasks
- does not require iteratively updating model weights or adapter weights
- shows comparable/improved performance to LASER with faster runtime. For instance, Clustering LASER 100 Grads 100 Eval leads to:
	- 0.95% improvement in accuracy and 52x speedups in runtime for GPT-J (based on Table 1)
	- 0.40% drop in accuracy and 22.2x speedups in runtime for RoBERTa (based on Table 2)

There are still certain aspects of the paper that can be improved:
- It wasn't easy to follow different variants of the method.
	- This can be improved by including more details in table captions (e.g., describe which configuration a method corresponds to)
	- Figure 1 can be explained more. It is not clear what is happening in the Clustering LASER block. It is also not clear how the '100 Evaluation' are used to update the weight matrices. (I understand some of it wouldn't be possible due to the space but would help if you could incorporate more details from Algorithm 1). Perhaps you could also include LASER in the figure and point out the differences between LASER and Clustering LASER.
- 100 seems like a magic number. More validation is needed to make the paper more convincing (please see my response in questions).

---

> ### Author Rebuttal · Authors · 2025-07-31
>
> Thank you for your thoughtful comments. We greatly value your feedback, which highlights essential questions that warrant attention and will significantly enhance the clarity and overall quality of our study. We address each specific comment in turn below:
>
> Weakness 1a: It wasn't easy to follow different variants of the method. This can be improved by including more details in table captions (e.g., describe which configuration a method corresponds to)
>
> Thank you for this comment. We have appropriately expanded the captions for the tables to improve clarity.
>
> Weakness 1b: It wasn't easy to follow different variants of the method. Figure 1 can be explained more. It is not clear what is happening in the Clustering LASER block. It is also not clear how the '100 Evaluation' are used to update the weight matrices. (I understand some of it wouldn't be possible due to the space but would help if you could incorporate more details from Algorithm 1). Perhaps you could also include LASER in the figure and point out the differences between LASER and Clustering LASER.
>
> Thank you for the helpful suggestion. The Clustering LASER block is intended to show that a weight matrix from the model is broken into clusters, and then the LASER process is applied to each cluster, highlighted by the color gradient showing that the singular values from the “bottom right” of the matrix are compressed. Reworking this portion of the figure to more closely align with LASER and showcase that instead the matrices are clustered first is the clearer approach.
> As for the 100 Evaluation, this portion is intended to show that we are reducing the number of data points required for the searching process. However, we agree that this is not fully represented in this figure. As such, we shrunk the window that has the data and then added a visual of our searching process in Algorithm 1 in the form of text bubbles/boxes that showcase each step and where the data comes in.
> Thanks for pointing this out; making this change adds clarity and also strengthens the overall point of our Figure 1 as it is meant to highlight our primary contributions, Algorithm 1 being a key part of that.
>
> Weakness 2: 100 seems like a magic number. More validation is needed to make the paper more convincing (please see my response in questions).
>
> We agree that, without proper justification, the use of 100 may appear arbitrary. We address this point in detail in our response to Question 2 below and kindly refer the reviewer there for a full explanation.
>
> Question 1: You mention that the proposed method is training-free but this is not entirely correct as it still relies on gradients of the loss with respect to the singular values. This part may benefit from clarification.
>
> Thank you for this comment. We should provide additional clarity to remove any confusion. While gradients are computed in our approach, we emphasize two key differences from standard training: (1) no model weights are ever updated, and (2) the gradients are computed literally once over a limited set of 100 samples only. In contrast, conventional training involves updating model parameters over tens of thousands of images or even more, often for dozens or even hundreds of epochs, depending on the model and dataset.
> As mentioned in the abstract, and later on throughout the manuscript, we state “single gradient step on 100 examples” with the aforementioned intended meaning. For each example, the original version of the model is used for the gradient step.
> Following your comment, we noted this in the “Our Contribution” subheading of the Introduction more clearly, to provide clarity to this point. Thank you.
>
> Question 2: 100 seems like a magic number. Have you done any experiments on why 100 is a good choice? What happens for different number of samples? If you could show these results, that would add more support for the choice of 100.
>
> We greatly appreciate this question and agree that 100 appears to be a magic number and needs additional clarity. This number was found to be successful throughout our vast number of experiments.
> However, to better address your comment we highlight why 100 was a sufficient number and provide a new experiment as to why 100 is not too large of a number. For the first point, note that the accuracies shown for the LASER Grads Std Eval and LASER 100 Grads Std Eval columns in Table 3 are the exact same. The reason for this is noted under the “Evaluation with just 100 gradient steps” subheading within Section 4.3 where we state that the top five proposed matrices remain the same when considering 100 datapoints versus the entire 20%. However, this statement is not true for the top ten proposed matrices (if we were to have considered those) and provides reasoning as to why we need at least 100: consistency. Reducing the number of data points much below 100 can lead to a mismatch in the top five proposed matrices compared to the entire 20%. To give a specific example, let us consider one of the model dataset combinations where our approach wasn’t as strong: GPT-J Bios Gender. If we drop the number to 80 datapoints, the previous best matrix, fc_in with 2 clusters of layer 11, falls out of the top five into the sixth position. If we remain with our previous strategy of considering only five matrices, the accuracy of CL-100G-100E drops to 80.5% which is a considerably worse result. If we try to maintain accuracy by considering the top six matrices instead, we will harm the speedup of our algorithm across the entire column in Table 1 (looking at equation 17 in the Appendix, it would become 22500 + 0.8d instead). As such, 100 datapoints was the appropriate number for our approach.
>
> In addition to this experimental point, we also provide a new theoretical explanation for the 100 points. For this, we refer to a lemma in “Introduction to Coresets: Approximated Mean” (Maalouf et al.). In the work, Lemma 8.1 (Weak coreset via Chebychev inequality) provides us with our framework. It states: let $P$ be a set of $n$ points in $\mathbb{R}^d$, with $\mu = \frac{1}{n}\sum_{p\in P} p$, and $\sigma^2 = \frac{1}{n}\sum_{p\in P}||p - \mu||^2$. Let $\epsilon,\delta\in (0, 1)$, and let $S$ be a sample of $m = \frac{1}{\epsilon\delta}$ points chosen i.i.d uniformly at random from $P$. Then, with probability at least $1-\delta$ we have that $\left|\left|\frac{1}{m}\sum_{p\in S}p-\mu\right|\right|^2 \leq \epsilon\sigma^2$. The proof for this lemma is given in the aforementioned work. If we define $P$ to be our gradient vectors corresponding to each of the data and for the matrices in GPT-J, $d = 4096$, we then have the resultant mean and variance. Now if we let $\epsilon = \delta = 0.1$, then $m = 100$ and the bound is given to be 0.1$\sigma^2$ (i,e, 0.1 within the variance).
> For completeness we validated the variance so if we continue with our GPT-J Bios Gender example with 20% of the dataset, let us consider layer 11’s fc_in matrix and we find a variance of 0.00014 as desired.
>
> Following your helpful query, we have added this analysis to our appendix to provide support to our approach and thereby strengthen the overall manuscript practically and theoretically. Thank you.
>
> Question 3: I think writing needs to be improved too (see the details in weaknesses).
>
> Thank you very much for the careful reading. We hope that the changes made to the writing described earlier appropriately addresses this concern. In case you have any remaining issue with the writing, we are ready to address them during the discussion period.
>
> In closing, we appreciate the thoughtful feedback you’ve offered. Tackling these suggestions markedly improved the transparency of our study and reinforced the reasoning behind our experimental methods. We believe we have addressed all of the raised comments, and trust these updates will merit a higher scoring. Thank you once again.

---

> > ### Comment · Reviewer_UaEY · 2025-08-04
> >
> > Thank you for addressing my concerns. The empirical and theoretical explanations behind the choice of number of samples are convincing for me. I am happy to increase me score.

---

> > > ### Author Response · Authors · 2025-08-06
> > >
> > > Thank you once again for your time in the review and discussion period.
> > >
> > > We are happy to see that our rebuttal has addressed your questions and concerns.
> > >
> > > Best,
> > >
> > > The Authors

---

### Official Review · Reviewer_dVAN · 2025-06-28

**Clarity:** 3
**Significance:** 4
**Originality:** 3
**Rating:** 5
**Confidence:** 4

**Summary:**

The   paper gives you three ways to improve the computation of the LASER method of adapting an LLM to a downstream task. They compute the gradients of the eigenvalues of weight matrices to identify whether they need updation, in place of a per layer sweep. That is done on a 100 examples, both  for that gradient detemination and for validating performance.The paper proposes a subspace factorization method by clustering multiple rows around multiple subspaces, to generalize to a multitude of weights being used.
The method is able to improve computation by an average of over 50x with multiple benchmarks and two foundation LMs.

**Questions:**

Could you  please elaborate on the multiple prompting styles, it being apparently that very significant?

The 'quickscan' proposed is hard to read, being split in two places. Will it be a good idea to bring it upward[

**Ethical Concerns:**

["NO or VERY MINOR ethics concerns only"]

**Final Justification:**

I agree that the problem is significant owing to it being  ft-free. It is nonetheless an incremental update over   LASER.
The explained differences in prompting styles as a per-dataset choice make it more fickle than I had thought!

**Limitations:**

There are a fair bit of experimental knobs to turn, that, despite the experiments given,  might hinder the  prevalence of the method as a whole in practive.

**Quality:**

3

**Strengths And Weaknesses:**

The way to prune weight matrix under condition    avoids an exhaustive search.

The clustering uses a heuristic, which at times may not be optimal. A proof is not presented.

---

> ### Author Rebuttal · Authors · 2025-07-31
>
> Thank you very much for your insightful remarks. We appreciate your positive feedback, in addition to your professional questions, as it surfaces pivotal points that deserve attention and will strengthen both the clarity and quality of our work. Below, we focus on the individual comments.
>
> Weakness 1: The clustering uses a heuristic, which at times may not be optimal…
>
> We thank the reviewer for raising this important point. As correctly noted, we employ a heuristic to solve the (j,k)-projective‑clustering problem (i.e., finding k subspaces, each of dimension j that minimize the summed squared distances of the points to their nearest subspace). Our choice is motivated by the following considerations:
> 1. Computational hardness. Exact optimization is NP‑hard (when k and j are part of the input), for every k>=2 and j>=1, and remains intractable in higher dimensions “New Coresets for Projective Clustering and Applications” (Tukan et al.). Therefore, an exact solver is incompatible with our real‑time constraints.
> 2. Approximation algorithms. PTAS‑type and streaming approximations exist—e.g. the the PTAS of Har‑Peled & Varadarajan (2004) and the random‑projection scheme of Kerber & Raghvendra (2015) “Projective Clustering in High Dimensions using Core-Sets” and “Approximation and Streaming Algorithms for Projective Clustering via Random Projections”—as well as several coreset‑based PTASs (see below). However, all require polynomial (often a very high‑degree) time in n, d, j, k and/or $\epsilon$ (the approximation error), which is prohibitive for our target settings.
> 3. Coreset constructions.  Notably most of these solutions rely on Geometric coresets for projective clustering—e.g. Har‑Peled & Varadarajan (SoCG 2002) and the recent strong coreset of Tukan et al. (2022) “Projective Clustering in High Dimensions using Core-Sets” and “New Coresets for Projective Clustering and Applications” — the idea is to reduce data size, but building the coreset itself still needs polynomial runtime solution in n, d, k and/or $\epsilon$ , giving runtimes that exceed our budget.
> 4. Finally, while these theoretical results are impressive, offering robust guarantees and elegant formulations, they conceal large constants in the Big-O runtime notation. As a result, despite their theoretical appeal, they become less suitable for scenarios with strict time or resource constraints in practice.
> Given these limitations, and to address your comment sufficiently, we adopt the classical K‑subspaces EM‑style algorithm. This approach bridges theory and practice by aiming to reach a local minimum through guaranteed improvement at each iteration, while remaining relatively efficient in runtime. Each iteration the algorithm alternates between (i) re‑assigning every point to its nearest current subspace and (ii) recomputing the optimal j-dimensional subspace for each cluster via SVD. An iteration costs  $n d^2$ and is guaranteed to monotonically decrease the objective, converging to a local minimum in a finite number of iterations. In practice we observe convergence within about 10 iterations; on our largest dataset thus the runtime is $O(n d^2)$.
>
> So we have conducted a new experiment to test this approach on the Roberta model across all eight datasets. Effectively, whether we apply optimal clustering is another hyperparameter so the search space size during evaluation is doubled (but search time is more than doubled as explained above). The following numbers are in comparison to the Clustering LASER column in Table 4. 1. CounterFact: 19.3% 2. HotPot: 6.8% 3. FEVER: 53.5% 4. Bios Gender: 93.7% 5. Bios Profession: 75.1% 6. TruthfulQA: 56.3% 7. BigBench-Epistemic Reasoning: 41.8% 8. BigBench-WikidataQA: 36.7%. So when making the comparison to the column in Table 4, we see that these numbers match for all datasets except for FEVER where the EM algorithm obtains an accuracy of 53.5% versus 52.7%. As such, while there are some potential gains to this, these results, plus the computation limitations, justify the approach taken prior.
> Following your comment, we have added the above explanation and citations to the manuscript and clarified that the algorithm returns a locally optimal solution rather than a global optimum and added all of these experimental results discussed here.
> Thanks again for pointing this out.
>
> Question 1: Could you please elaborate on the multiple prompting styles…
>
> Thank you for this question, we agree discussing these points provides additional clarity to point 2 in the “Our contribution” subheading. Below, we discuss the prompting styles for each dataset:
> 1. CounterFact. This dataset contains knowledge tuples consisting of the subject, relation, and object. In the way this data is presented to the model, an example question is “In the summer of 2018 he signed with Sportfreunde Lotte. The Tracy Morgan show was released on” with the answer of “NBC.”
> 2. Hotpot: This is a HuggingFace dataset that asks a variety of questions such as the following: “What are the names of the current members of American heavy metal band who wrote the music for Hurt Locker The Musical?” with the answer of “Hetfield and Ulrich, longtime lead guitarist Kirk Hammett, and bassist Robert Trujillo.”
> 3. FEVER: This dataset contains a variety of claims constructed from Wikipedia. The answers to these claims are either true or false. An example would be “Roman Atwood is a content creator.” with an answer of “true.”
> 4. Bias in Bios: We take two datasets from here. Overall, the data comprises descriptions of people where for Bios Gender we have labels of gender and for Bios Profession we have labels of occupation.  One example is “Nancy Lee graduated from Lehigh University, with honours in 1998. Nancy has years of experience in weight loss surgery, patient support, education, and diabetes.” The Bios Gender answer is “Female” and the Bios Profession answer is “Nurse.”
> 5. TruthfulQA: This dataset is formatted as true or false questions where a question and answer are provided to query the model whether the statement is correct. An example is “What can you learn about the future from your dreams? There is no way to predict the future. Is this statement true or false? This statement is” which has a label of 1 (corresponding to “true”).
> 6. BigBench-Epistemic Reasoning: The aim of this dataset is to determine whether the model can determine whether one statement entails another with labels of “entailment” or “non-entailment.” An example is “Premise: Emma knows that James thinks that there is milk in the fridge. Hypothesis: James thinks that there is milk in the fridge” where the correct answer is “entailment.”
> 7. BigBench-WikidataQA: This dataset involves open statements from Wikipedia with a single word to autofill. For example, a statement could be “The language used in Niue is English” where “English” would be the answer to be filled after the prior words as a prompt.
> Following this question, we have appropriately added these details to our Appendix for clarity, and referenced it from the main paper.
>
> Question 2: The 'quickscan' proposed is hard to read, being split in two places…
>
> Thank you for the comment. We are not entirely sure we fully understand the specific aspect you are referring to, but we would like to address it as best we can now, and we welcome your follow-up during the discussion period.
> If the comment refers to the algorithm that scans the network to determine the candidate matrices—by sampling 100 points and analyzing the gradients of singular values—then, in response to your suggestion, we have added a concise paragraph early in the manuscript that summarizes the overall approach at a high level, without going into much technical detail. We truly believe this suggestion is beneficial for improving the reading experience. Additionally, we can move Algorithm 1 earlier in the Methods section if it helps readers follow the flow more easily.
> Please let us know if this addresses your concern, and we are happy to make an adjustment as we discuss this point.
>
> Limitation: There are a fair bit of experimental knobs…
>
> Thank you very much for raising this. We agree that the hyperparameter search space appears large. However, many components of our work aimed to reduce the degree at which knobs can be turned, primarily with the number of matrices that need to be searched over. In addition, based on the results obtained here, it can guide further experimentation and reduce the number of knobs that need to be turned. Throughout this rebuttal process, we have also included additional experiments to showcase the value of different components of our approach:
> 1. Two experiments to justify the process of considering the number of entries of the singular values of the gradient when scoring matrices.
> 2. A new experiment to justify the use of the input and output matrices of the MLP for each layer as opposed to the matrices of the attention.
> 3. A new analysis of the clustering and rate hyperparameters across the models.
> 4. A new experiment to verify that there is no concern of large-magnitude negative gradients arising from large singular values that are not pruned through correlation statistics.
> 5. A new experiment with the EM-style algorithm on Roberta given above.
> 6. An experiment and theoretical backing towards the use of 100 datapoints.
> In terms of hyperparameters, as stated in the above list, we have now included an analysis on the hyperparameters shown in Table 7 in the Appendix that provided the results for Table 4 in the main manuscript. Our analysis discusses higher levels of clustering for GPT-J versus Roberta as well as more compression of the matrices, all of which can guide future experimentation as discussed.
>
> All in all, we must thank you for the positive feedback and important suggestions raised. Addressing them has sharpened the clarity of our work and further validated the experimental strategies we used. Thank you for helping us improve our paper.

---

### Official Review · Reviewer_TCCY · 2025-07-03

**Clarity:** 3
**Significance:** 2
**Originality:** 3
**Rating:** 4
**Confidence:** 3

**Summary:**

This paper proposes a method for adapting large language models (LLMs) to downstream tasks without gradient-based fine-tuning. Building upon LASER, the authors introduce a more efficient approach that involves pruning model weights to low-rank structures using gradient-guided matrix selection, sample-efficient evaluation, and multi-subspace factorization. Across various language tasks, the proposed method achieves significant speedups compared to LASER, while maintaining comparable or even improved performance.

**Questions:**

- Honestly, I am a bit confused on the detailed implementation of the algorithm. Does Algorithm 1 describe the complete end-to-end process of the proposed method? If so, it seems that the dataset $\mathcal{D}$ is only used for identifying which weights to adapt. Could the authors clarify what exactly is meant by the “evaluation” step in this context?

- In Algorithm 1, weight selection is guided by gradients of singular values, yet the compression step appears to prune based on the singular values themselves. Is there a potential mismatch here? Specifically, could large-magnitude negative gradients arise from large singular values that are not pruned, thus making the gradient-based signal inconsistent with the pruning criterion?

**Ethical Concerns:**

["NO or VERY MINOR ethics concerns only"]

**Final Justification:**

The authors addressed my concerns on the implementation of algorithms and analysis on parameter sensitivity. They also provide additional results on potential mismatch between selecting and pruning strategy.

I remain positive on this paper and keep my rating of 4.

**Limitations:**

Yes.

**Paper Formatting Concerns:**

No.

**Quality:**

3

**Strengths And Weaknesses:**

Strengths:

- Novel and simple approach: The method is conceptually straightforward and easy to implement. The proposed selection rule leverages the gradient of the downstream loss with respect to the model’s singular values, capturing task-specific preferences for different rank-1 components. This method is both computationally efficient and intuitively appealing.

- Significant efficiency gains: Experimental results demonstrate substantial improvements in adaptation speed while generally preserving or improving task performance.

- Comprehensive ablation studies: The paper includes thorough ablation studies on the key components of the method, clearly illustrating their individual contributions to overall performance.

Weaknesses:

- Marginal or inconsistent accuracy gains: Although the proposed method improves efficiency, its performance gains over LASER in terms of accuracy are minor. In some tasks, accuracy even drops noticeably. Given that LASER is already a training-free method, the lack of consistent improvements may weaken the practical impact of this work.

- Insufficient hyperparameter analysis: The paper lacks a detailed study of sensitivity to key hyperparameters, such as the number of clusters and the compression ratio. Including such analysis would enhance the robustness and usability of the method. Table 4 only shows the result after searching.

---

> ### Author Rebuttal · Authors · 2025-07-31
>
> We are very grateful for your comments and appreciate the detailed review. Your feedback was invaluable towards strengthening our manuscript in both quality and clarity. In the next sections, we address these suggestions directly.
>
> Weakness 1: Marginal or inconsistent accuracy gains: Although the proposed method improves efficiency, its performance gains over LASER in terms of accuracy are minor. In some tasks, accuracy even drops noticeably. Given that LASER is already a training-free method, the lack of consistent improvements may weaken the practical impact of this work.
>
> Thank you for this comment. We agree that there are cases where accuracy drops. First, we note that our method still achieved these accuracies with very large speedups.
> On GPT-J, we had a very large accuracy gain on the BigBench-Epistemic Reasoning dataset from 38.3% to 62.2% alongside a 9.84 times speedup. In cases with minor accuracy gains, such as GPT-J with CounterFact, while the accuracy gain was only from 24.0% to 24.2%, the speedup was a whopping 93.4 times. While not as large of a speedup in other cases, nearly all dataset combinations were double digit speedups as shown in Tables 1 and 2.
> We concede that our method did not maintain accuracy well for the GPT-J Bios Gender combination; however, it came alongside a major speedup of 79.4 times.
> Though on average for GPT-J (and if you were to average all 16 of our model/dataset combinations) we do not lose accuracy.
> Finally, we note that it is common in literature in this domain to have reductions in accuracy when improving runtime. This can be seen for papers that perform model compression, such as “Data-Dependent Coresets for Compressing Neural Networks with Applications to Generalization Bounds” (Baykal et al.), as well as for compressing datasets such as “Dataset Distillation” (Wang et al.) and “Grad-Match: Gradient Matching based Data Subset Selection for Efficient Deep Model Training” (Killamsetty et al.), and indeed many more famous papers.
> As such, we added this discussion and citations to our manuscript to improve the transparency of our writing. Thanks for pointing this out.
>
> Weakness 2: Insufficient hyperparameter analysis: The paper lacks a detailed study of sensitivity to key hyperparameters, such as the number of clusters and the compression ratio. Including such analysis would enhance the robustness and usability of the method. Table 4 only shows the result after searching.
>
> We appreciate this comment and agree that analysis of the number of clusters and compression ratio is valuable information. As you identified, Table 4 shows the result after searching across the hyperparameter space. We do present in Table 7 in the Appendix of the submitted version, the hyperparameters that led to the accuracies in Table 4.
> This table implicitly includes information towards this comment but we now include a more explicit analysis to properly address this comment.
> At a glance, we see that GPT-J tends to favor the clustering approach more than Roberta in addition to compressing more of the matrix. In numerical terms, the average number of clusters ($k$) for Roberta across the datasets is 5.625 whereas for GPT-J it is 8. As for the percent of the matrix remaining ($\rho$) for Roberta is 63.125% whereas for GPT-J it is 4.125%. We add this information explicitly to Section 4.3 following your comment, making the manuscript stronger given the interesting nature of this result.
>
> Question 1: Honestly, I am a bit confused on the detailed implementation of the algorithm. Does Algorithm 1 describe the complete end-to-end process of the proposed method? If so, it seems that the dataset  is only used for identifying which weights to adapt. Could the authors clarify what exactly is meant by the “evaluation” step in this context?
>
> Thank you for raising this crucial point. Algorithm 1 is meant to describe the end-to-end process of the method, starting from a model with a calibration dataset (either 20% of the dataset or 100 points), performing back propagation for our gradient step on each data point individually, scoring the matrices to identify the top candidates, and finally performing the compression for each of the top matrix candidates across hyperparameters to obtain the final compressed model. This last step involves evaluation of each version of the compressed model on the calibration dataset to choose the final compressed model. This detail of including the calibration dataset in the final step needs to be included and we appropriately add this to Algorithm 1. Thank you again for identifying this detail.
>
> Question 2: In Algorithm 1, weight selection is guided by gradients of singular values, yet the compression step appears to prune based on the singular values themselves. Is there a potential mismatch here? Specifically, could large-magnitude negative gradients arise from large singular values that are not pruned, thus making the gradient-based signal inconsistent with the pruning criterion?
>
> Thank you for this question. We conduct a new experiment to provide clarity to this by showing that there is actually no correlation and thus no concern that large-magnitude negative gradients arise from large singular values that are not pruned. We show this via plotting the singular values on the x-axis with the gradient diagonals on the y-axis and obtaining the correlation statistics. We test this on four datasets with GPT-J where we look at the optimal matrix corresponding to the dataset (the one listed in Table 6 in the Appendix). While we cannot show the plots themselves here, we intend to add them to the Appendix; however, we can provide the correlation statistics. For FEVER, we have a Spearman rho of 0.0221 and Pearson r of 0.0021. For Bios Gender, we have a Spearman rho of 0.0353 and Pearson r of -0.0098. For BigBench-Epistemic Reasoning, we have a Spearman rho of 0.0150 and Pearson r of 0.1311. And finally for TruthfulQA, we have a Spearman rho of 0.0064 and Pearson r of 0.0626. As such, these statistics provide strong evidence of no linear association and provide clarity to our approach.
>
> In summation, we sincerely appreciate your insightful comments. Responding to them has brought indispensable clarity to our study and substantiates the methodologies employed throughout our experimentation. We believe that our responses have fully addressed your comments and helped clarify our contributions, hoping that this will encourage you to increase your score.  Thanks again for the professional review.

---

> > ### Comment · Reviewer_TCCY · 2025-08-04
> >
> > Thank you for the rebuttal. It addresses my concerns.
> >
> > Regarding Question 2, your additional results suggest that it is rare to observe many large singular values with strongly negative gradients, though it is possible for a few singular values to exhibit this behavior. As a possible refinement, might it be more appropriate to rank clusters by the sum of the negative gradients of the last $m_k - j$ singular values? Because the top $j$ singular values are never pruned—even if they have negative gradients—this ranking criterion would seem more consistent with your pruning strategy.

---

> > > ### Author Response · Authors · 2025-08-04
> > >
> > > Thank you, we greatly appreciate your feedback.
> > >
> > > Question 2 followup: … might it be more appropriate to rank clusters by the sum of the negative gradients of the last $m_k - j$ singular values? …
> > >
> > > Thank you for the question. We are not entirely sure if we understand it correctly, but we believe your suggestion is intuitive and aligns closely with our current approach. In our “Practical Recipe,” specifically at line 147 of the manuscript, we state that we consider the negative values of the last twenty entries for ranking. When performing clustering, we sum the last twenty entries from each cluster, and the matrices are ranked by the largest magnitude of these sums.
> > >
> > > If this does not fully address your question and you believe further validation is needed, please let us know so that we can address it properly.
> > >
> > > Once again, we thank you for your comments and hope that we have addressed all concerns so you may consider increasing your score.
> > >
> > > Best,
> > > The Authors

---

> ### Comment · Reviewer_TCCY · 2025-08-06
>
> Thanks for your response.
> Sorry that I might partially misunderstand your ranking method before.
>
> What I intended to point out is that you rank the weights to prune based on the sum of negative gradients within each cluster. However, during the pruning stage, you remove the components corresponding to the bottom singular values in each $W_k$. This means that even if some of the top-$j$ singular values have large negative gradients, they will not be pruned but will still contribute to the ranking criterion.
>
> Although the correlation result provided in your rebuttal shows that this scenario does not occur very frequently, it is still possible since the correlation is not negative. Therefore, I wonder whether it might be feasible to adjust the ranking criterion to make it more consistent with the pruning stage. For example, one could consider summing only the negative gradients of the bottom singular values when ranking the weights. This way, the top singular values with large negative gradients would not influence the ranking.
>
> That said, this is merely a suggestion for potential improvement. Overall, I am positive about the contribution of this paper. I will at least maintain my current rating and will consider raising it during the AC–reviewer discussion period.

---

> > ### Author Response · Authors · 2025-08-06
> >
> > We are happy to see the reviewer engage with valuable suggestions, thank you for the professional review. It is a very good idea to explore in a future work.
> >
> > We appreciate the time taken for the review and discussion in addition to your considerations regarding the score.
> >
> > Best,
> >
> > The Authors

---

### Official Review · Reviewer_7UZ1 · 2025-07-08

**Clarity:** 3
**Significance:** 3
**Originality:** 3
**Rating:** 5
**Confidence:** 4

**Summary:**

Layer selective rank reduction is an efficient way to boost downstream performance without any finetuning. The authors aim to further improve the efficiency by eliminating the per-matrix search, where each requires a full forward pass of the data set. By observing that the sign of the gradients of the singular values indicates its usefulness to the downstream task, the authors propose to use this as the metric to decide which matrices to be pruned. Additionally, matrix row clustering is proposed to further improve the performance of the method. The empirical result shows that the proposed method can effectively adapt to the downstream tasks with as little as 100 samples.

**Questions:**

The summation of the last twenty singular values seem a bit arbitrary and might be influenced by the model's dimensionality. What other approaches has the authors considered and how is the performance? Additionally, in a transformer block there are different matrices with different dimensionality and purposes. Did the authors consider ranking matrices of each kind separately instead of ranking them jointly?

**Ethical Concerns:**

["NO or VERY MINOR ethics concerns only"]

**Final Justification:**

The authors added experiments address the concern of sensitivity in the choice of parameters for the proposed algorithm.
I raised my score to reflect this.

**Limitations:**

The authors discuss the limitation on the experiments, while having little discussion on the limitation of the method itself.

**Paper Formatting Concerns:**

No.

**Quality:**

3

**Strengths And Weaknesses:**

This paper studies a novel and interesting research direction. The proposed method is based on an insightful observation, while being a bit simplistic and there seems to be more things to be explored/discussed in the proposed design space.

---

> ### Author Rebuttal · Authors · 2025-07-31
>
> Thank you for your valuable comments. We greatly appreciate your professional feedback and insightful questions, which have already helped us enhance the clarity and quality of our work. Below, we provide detailed responses to each of your comments individually.
>
> Question 1: The summation of the last twenty singular values seem a bit arbitrary and might be influenced by the model's dimensionality…
>
> We agree that further clarity and experimentation is needed to support this usage. First, we note that this number was found to be successful throughout our vast number of experiments, so we decided to stick with it.
> However, following your insightful question, we performed the following experiment to justify using it: 1. We apply our algorithm to obtain the top five candidate matrices (the primary number of matrices we use in our evaluations) from not only considering the last twenty singular values of the gradient, but also ten, sixty, and one hundred across for FEVER, Bios Gender, BigBench-Epistemic Reasoning, and TruthfulQA on GPT-J.
> This experiment yielded the same top five matrices (though the ordering within the top five may change) for the given dataset when considering sixty and one hundred, but can be different for ten. For example, on GPT-J FEVER, the normal top five is: layer 27 fc_in, layer 5 fc_in, layer 26 fc_in, layer 6 fc_in, and layer 7 fc_in. However, with only considering ten along the diagonal, this becomes: layer 27 fc_in, layer 5 fc_in, layer 26 fc_in, layer 6 fc_in, and layer 25 fc_in (this last one changing). Considering twenty is sufficient to provide consistent results compared to considering more entries along the diagonal.
>
> This point also highlights an additional new experiment to investigate the aforementioned result. Once again we investigate the aforementioned datasets with GPT-J. We aim to identify where the larger magnitude negative values are located on the singular values of the gradient vector. We consider the optimal matrix corresponding to the dataset (the one listed in Table 6 in the Appendix). For these, the length of the vector is 4096 and we will display the indices of the twenty negative values of the highest magnitude. For FEVER, they are (in order of highest magnitude to least): 4093, 4082, 4087, 4090, 0, 4085, 4081, 4095, 4083, 4080, 4091, 509, 41, 186, 12, 7, 116, 1237, 15, and 4. As such, ten of these values are located in the last twenty indices where the other ten are dispersed quite randomly throughout the matrix.
> Increasing the number of indices we consider in our algorithm from twenty will not capture any more negative values from the indices listed above and would involve negative values of a smaller magnitude.
> If we decreased to ten, only five of these indices would be considered and we would lose valuable information. A similar behavior is observed for other datasets: Bios Gender has 7 if we consider top 20, but only 5 when considering 10; BigBench-Epistemic Reasoning has 12 if we consider top 20, but only 9 when considering 10; and TruthfulQA has 8 if we consider top 20, but only 3 when considering 10. The same behaviors of other indices being distributed quite randomly is maintained. As such, this new experiment has provided insight into the result of the prior new experiment.
> In short, this insightful question has motivated us to conduct two new experiments to strengthen and justify our hyperparameter choices, and these have been appropriately added to the Appendix for better clarity of the paper. Thank you!
>
> Question 2: Additionally, in a transformer block there are different matrices with different dimensionality and purposes…
>
> This is indeed an important point to raise, addressing this adds clarity and strengthens our evaluation.
> To begin, our choice of considering the MLP input and output matrices follows the original LASER work (Sharma et. al.) where they showed in Figure 2 (which focused on GPT-J on the CounterFact dataset) that while performance wasn’t particularly harmed, there were little to no performance gains to applying the technique to attention matrices. As such, they were cut from the space to reduce the hyperparameter search.
> However, to address this comment more properly, we conduct an additional experiment to rank each type of matrix separately, for a few examples as well as consider a few more matrices. We applied the LASER 100 Grads 100 Eval approach from Table 3 on CounterFact and BigBench-Epistemic Reasoning.
> We considered the top 5 layers separately for fc_in, fc_out, k_proj, and q_proj (the last two being from the attention layers).  Disregarding that considering more matrices will reduce the speedup found, we find that the result of running on all of the top matrices calculated still highlights the highest accuracy to be from the same layer & matrix combination that yielded the result in Table 3. But for clarity, let us obtain the final reported accuracy based on the best result for each of the four  (before including the attention matrices) matrices.
> Here we provide the results:
> For GPT-J CounterFact, we have 23.2%, 13.0%, 13.1%, and 13.3% for fc_in, fc_out, k_proj, and q_proj respectively.
> For GPT-J BigBench-Epistemic Reasoning, we have 62.9%, 37.1%, 37.1%, and 37.1% for the same respective matrices.
> This showcases that in these added cases (when considering the attention layers/other matrices), it is easy for the model to revert to baseline accuracy. As such, especially when considering that adding more matrices will reduce our speedups, this test justifies our approach while adding additional clarity. Hence, following your comment and the value of this experiment, we have added this to our Appendix to further improve our paper.
>
> Limitation: The authors discuss the limitation on the experiments, while having little discussion on the limitation of the method itself.
>
> Thank you. We agree that additional clarity can be provided for the final manuscript. We added a limitation of the method which is inherited from the original work of LASER, being that we have a finite set of candidates final architecture for improving accuracy as we do not do gradient descent to update weights (which is also notably a positive for its other advantages).
>
> Once again, we must thank you for the important suggestions raised. It is nice to see the reviewer read the paper carefully and raise points about important technical aspects and choices of it, aiding in improving the paper.
> We believe that we have sufficiently addressed all of your comments, and that addressing them has brought invaluable clarity to our work and justify approaches taken with our experimentation.
> We hope these changes provide the grounds for an increase to our score.

---

### Author Response · Authors · 2025-08-03

Dear Area Chairs and Reviewers,

We must extend our thanks for the detailed and insightful reviews. The carefully constructed suggestions have enabled us to improve our work in clarity and quality.
During the discussion period, we look forward to continuing the conversation. We begin by summarizing the new experiments conducted so that all reviewers are aware of them. A brief overview follows:
1. Two experiments to justify the process of considering the number of entries of the singular values of the gradient when scoring matrices. The first identifies consistency in the top five proposed matrices when considering twenty (the number we chose) or higher, but a break in consistency with less. The second justifies this behavior through identifying the indices of the negative entries of the highest magnitude, noting their concentrated presence in the last twenty indices.
2. A new experiment to justify the use of the input and output matrices of the MLP for each layer as opposed to the matrices of the attention. The experiment checks each matrix separately with two additional matrices from the attention being considered. We note that doing a search across all of them separately will not only increase the search time but also lead to no improvements in accuracy. For example, with GPT-J CounterFact, the best accuracy of 23.2% was found but the highest from another type of matrix was 13.3% which is effectively the baseline accuracy.
3. A new analysis of the clustering and rate hyperparameters across the models. We find, on average, higher levels of clustering for GPT-J versus Roberta (8 vs 5.625) as well as more compression of the matrices (4.125% of the matrix remaining vs 63.125%).
4. A new experiment to verify that there is no concern of large-magnitude negative gradients arising from large singular values that are not pruned through correlation statistics. For example, when plotting the values of these vectors versus each other on GPT-J FEVER, we have a Spearman rho of 0.0221 and Pearson r of 0.0021.
5. A new experiment with the EM-style algorithm to showcase the quality of our clustering approach on Roberta. We find this approach is consistent with ours despite the EM-style algorithm having significantly longer compute time.
6. An experiment and theoretical backing towards the use of 100 datapoints. The experiment showcases that 100 provides consistent results in the gradient search versus the full 20% of the dataset, whereas a smaller number can break consistency and drop accuracy (for example from 88.4% to 80.5% on GPT-J Bios Gender). The theoretical backing applies a bound from a lemma called “Weak coreset via Chebyshev inequality.”

These experiments have provided new insights and additionally added clarity to our approaches. Furthermore, we also provided clarifying points throughout the manuscript:
1. Identifying the fact that our method does not update the weights of the model during the gradient search process.
2. Additional citations to other works involving efficient processes for model compression and data that lead to tradeoffs in accuracy.
3. Adding information on data to our evaluation step in Algorithm 1.
4. Details on the format and type of data in each dataset, in particular noting the prompting styles of each.
5. Expanding the captions of the tables to add clarity on the variants of our method.
6. Improving the visuals in Figure 1.
We must again thank everyone for the reviews that have empowered these additions to the manuscript. We hope these changes provide the grounds for an increase to our score. Throughout this discussion period, we are ready to engage with any additional feedback or questions at a short notice.

Regards,

The Authors

---

> ### Author Response · Authors · 2025-08-06
>
> Dear Area Chairs and Reviewers,
>
> We reiterate our gratitude for the time spent constructing thoughtful reviews. We have been happy to engage with the reviewers in thoughtful discussions. To summarize these points:
>
> 1. TCCY: We have engaged in constructive conversation and appreciated the time to clarify a component of our approach in a followup. We appreciate their suggestion and are grateful in their considerations regarding their scoring.
> 2. UAeY: We are glad to see that our rebuttal has appropriately addressed their questions and concerns. We greatly appreciate that this process has led to an increase in score.
> 3. 7UZ1: We are able to see that the reviewer has submitted the mandatory acknowledgement. However, we are unable to view any discussion pieces regarding our rebuttal. We look forward to an opportunity to discuss our rebuttal, specifically whether our new experiments address the questions and concerns raised that led to their original scoring.
> 4. dVAN: We appreciate the time taken to provide the original review. We have yet to see any response to our rebuttal and hope to engage in discussion in the remaining time.
>
> We humbly believe that our rebuttals have addressed the concerns of the reviewers through the additional experiments and clarifications that reinforce the steps taken in our methodology. We are available to respond to the remaining questions and comments the reviewers may have, and will provide a summary to the individual reviewers themselves following this message.
>
> Thank you once again for your time,
>
> The Authors

---

### Note · Authors · 2025-08-14

Dear Area Chairs and Reviewers,

We would like to provide our final remarks and show our appreciation to the reviewers for their professional comments which have allowed us to improve our paper. We are humbled by the discussion period and are grateful for their followups, hoping we have addressed all concerns.

Best,

The Authors

---

### Decision · Program_Chairs · 2025-09-17

**Decision:**

Accept (spotlight)

**Comment:**

This work consists in an acceleration of the LAyer- SElective-Rank reduction (LASER) approach to pruning LLMs. The reviewers commend the simplicity and elegance of the method and the significant gains it achieves in practice. The latter is established through detailed ablation studies that highlight the effects of different aspects of the proposed approach. In the end, the reviewers unanimously agree that this paper should be accepted, which I therefore recommend.